# Evaluating Cognitive Maps and planning in Large Language Models with CogEval

**Ida Momennejad**[*]
Microsoft Research
New York, NY
`idamo`

**Hosein Hasanbeig**[*]
Microsoft Research
New York, NY
`hosein.hasanbeig`

**Felipe Vieira Frujeri**[*]
Microsoft
Redmond, WA
`felipe.frujeri`

**Hiteshi Sharma**
Microsoft
Redmond, WA
`hiteshi.sharma`

**Robert Osazuwa Ness**
Microsoft Research
Redmond, WA
`robertness`

**Nebojsa Jojic**
Microsoft Research
Redmond, WA
`jojic`

**Hamid Palangi**
Microsoft Research
Redmond, WA
`hpalangi`

**Jonathan Larson**
Microsoft Research
Redmond, WA
`jolarso`

`@microsoft.com`

## Abstract

Recently an influx of studies claims emergent cognitive abilities in large language models (LLMs). Yet, most rely on anecdotes, overlook contamination of training sets, or lack systematic Evaluation involving multiple tasks, control conditions, multiple iterations, and statistical robustness tests. Here we make two major contributions. First, we propose CogEval, a cognitive science-inspired protocol for the systematic evaluation of cognitive capacities in LLMs. The CogEval protocol can be followed for the evaluation of various abilities. Second, here we follow CogEval to systematically evaluate *cognitive maps* and *planning ability* across eight LLMs (OpenAI GPT-4, GPT-3.5-turbo-175B, davinci-003-175B, Google Bard, Cohere-xlarge-52.4B, Anthropic Claude-1-52B, LLaMA-13B, and Alpaca-7B). We base our task prompts on human experiments, which offer both established construct validity for evaluating planning, and are absent from LLM training sets. We find that, while LLMs show apparent competence in a few planning tasks with simpler structures, systematic evaluation reveals striking failure modes in planning tasks, including hallucinations of invalid trajectories and falling in loops. These findings do not support the idea of emergent out-of-the-box planning ability in LLMs. This could be because LLMs do not understand the latent relational structures underlying planning problems, known as cognitive maps, and fail at unrolling goal-directed trajectories based on the underlying structure. Implications for application and future directions are discussed.

## 1 Introduction

Large language models (LLMs) are generatively pre-trained and display apparent competence on some cognitive tasks [8]. This has led to a recent surge in studies claiming LLMs have emergent human-level cognitive abilities, which may encourage applications that interact with LLMs in a zero-shot or few-shot manner with expectations of human-level cognition. However, most claims of competence are based on anecdotes rather than systematic evaluation. In response, we make two contributions. First, we propose CogEval, a Cognitive Science-Inspired [12, 5, 39] protocol for Measurement and Evaluation of cognitive abilities in LLMs (Figure 1, top), such as planning, theory

---

[*]Equal contribution

of mind, causal inference, or other abilities. Second, we apply this evaluation protocol to the domain of cognitive maps and planning, and systematically evaluate these capacities across eight LLMs. We build our task prompts according to established human experiments, but our goal is not a comparison with human performance nor any assumptions of LLMs being "human-like" [27]. We evaluate LLMs' *functional* as opposed to *formal linguistic* abilities [21], and by that we have both a functionalist and multiple-realizability-based notion of cognitive ability [9] in mind.

We investigated whether LLMs (OpenAI GPT-4, GPT-3.5-175B, and davinci-003-175B, Google Bard, Cohere-52.4B, Anthropic Claude-1-52B, LLaMA-13B, and Alpaca-7B) understand the latent structure of planning problems (cognitive maps). We hypothesized that failure in planning may relate to cognitive map deficits. To address these questions, we followed the CogEval protocol (Figure 1). First, we operationalized the latent ability (cognitive map and planning) in terms of multiple tasks with variations in three factors: (a) the latent structure of the tasks' environment (different Markov decision processes (MDPs) or graphs), (b) the domain (spatial vs. social ties vs. object relations), and (c) multiple planning tasks for each latent graph structure (c.f. Section 2 for detail). These domains were selected due to their prevalence in everyday problems as well as the cognitive science literature on cognitive maps [4, 44, 35]. We then generated repeated measurements across small and large LLMs (c.f. Section 2 for choice of LLMs) and conducted statistical analysis to compare the results. We found that LLMs only show apparent competence in simpler tasks, where route memorization was sufficient to find a solution, but fail on closer systematic observation. Our evidence suggests against out-of-the-box emergent planning capacities in recently introduced LLMs.

*What is a cognitive map?* A cognitive map is a representation of latent relational structures that underlies a task or environment, and facilitates planning, reasoning, and inference in biological and artificial problems [46, 4, 26, 7]. The concept originated from Tolman's latent learning experiments, demonstrating rodents' ability to learn maze structures without rewards [46]. This challenged the dominant behaviorist view that learning only occurs with reinforcement; and paved the way for a cognitive revolution. Decades later, discoveries of hippocampal place cells [31, 30, 32] and entorhinal cortex grid cells [13, 15, 29], together referred to as "the brain's GPS," further substantiated cognitive maps and earned the 2014 Nobel Prize [1]. Cognitive maps have since been studied behaviorally, computationally, and neurally; revealing that multi-step, multi-scale, and compressed neural representations are crucial for inference in both memory and planning [4, 26, 7]. Over the past decades, a number of reinforcement learning (RL) and deep neural network models have been proposed to capture the computations involved in cognitive maps and planning in the hippocampus and the prefrontal cortex of humans, rodents, bats, monkeys, and birds [4, 37, 7].

*Why would LLMs plan with a cognitive map?* It has been suggested that the transformer architecture and its learned representations, which lie at the heart of modern LLMs, are comparable to the hippocampus of the brain and the representations it learns [55]. Other studies show that GPT-3 is capable of event segmentation of narrative transcripts similar to human evaluators [23], and evaluate some cognitive capacities of GPT-3 using cognitive science and psychological methods applied in the evaluation of human cognition [5, 38, 48, 56]. Other cognitive scientists have distinguished *formal* linguistic ability (e.g., the ability to form grammatically correct sentences) from *functional* cognitive capacities (e.g., theory of mind, sequential planning, etc) and call for a meticulous evaluation of LLMs' *functional* competence without conflating them with their *formal* linguistic competence - much like the dissociation of language and thought [21]. Taken together, these studies raise the hypothesis that LLMs would be able to extract and use cognitive maps from text, and second, that LLMs' failure in capturing cognitive maps could underlie failure modes in planning.

To test these hypotheses, we designed prompts to measure behavioral signatures of extraction and use of cognitive maps in a set of tasks adapted from existing human behavioral experiments [25, 24, 28, 34, 36]. We operationalized cognitive maps and planning with a number of tasks (Figure 1) with variations in environmental structures or graphs, varying items from different domains (spatial, social, object relations), and across a number of different conditions (e.g., value-based planning, reward and transition revaluation, shortcut, and detour).

Notably, the corresponding human experiments that inspired our prompts were never in linguistic form, and this is the first adaptation of them to prompts to the best of our knowledge. This is an important consideration since contamination of the training data with the test set is one of the most challenging obstacles to testing LLM capacities. To prevent any possible contamination, we avoided BIG-bench [40], which has been flagged by OpenAI for contamination [2], and a planning benchmark

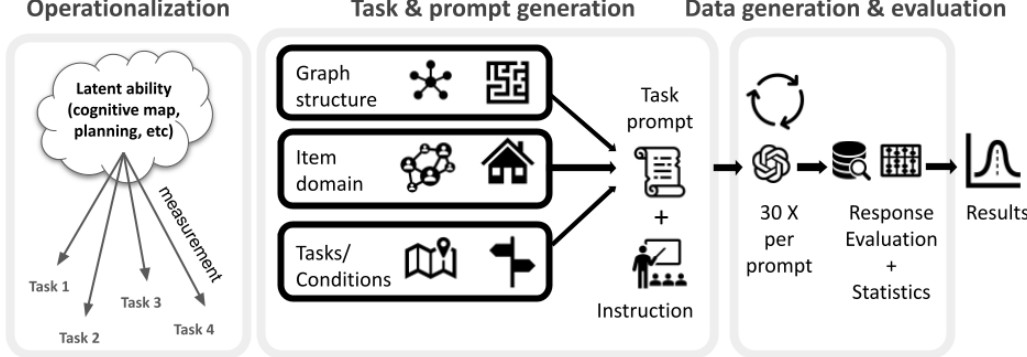

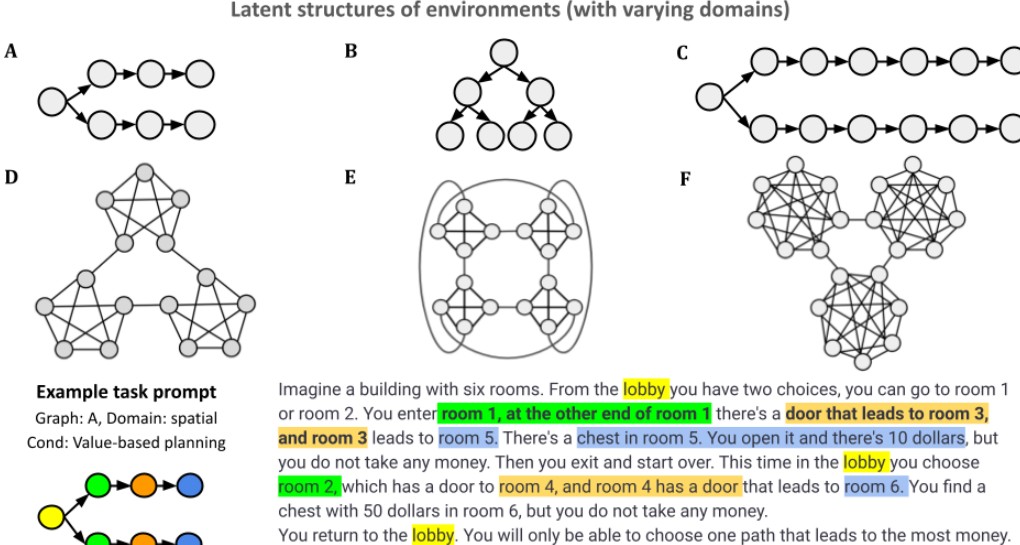

**Example task prompt**
Graph: A, Domain: spatial
Cond: Value-based planning

Imagine a building with six rooms. From the lobby you have two choices, you can go to room 1 or room 2. You enter room 1, at the other end of room 1 there's a door that leads to room 3, and room 3 leads to room 5. There's a chest in room 5. You open it and there's 10 dollars, but you do not take any money. Then you exit and start over. This time in the lobby you choose room 2, which has a door to room 4, and room 4 has a door that leads to room 6. You find a chest with 50 dollars in room 6, but you do not take any money.
You return to the lobby. You will only be able to choose one path that leads to the most money. Which room from the lobby will lead to the path where one can make the most money?

Figure 1: **The CogEval protocol, task structure, and example task prompt.** (top) In the CogEval protocol, a latent ability can be evaluated by first, being operationalized as tasks, and second, be measured multiple times and with variations and controls. We followed this protocol to evaluate cognitive map and planning. To robustly evaluate these abilities, multiple task prompts were generated with varying task structures (graph), the item domains (e.g., spatial or social), and task conditions (e.g., value-based path, detour). LLM responses were generated 30 times per task prompt and temperature for the three OpenAI models studied in this work and once per task and temperature for other LLMs. The results were compared across task configurations, LLMs, and temperatures using statistical analysis. (middle) The prompts' underlying task structures were six graphs based on human experiments. A: simple line graph from [25]. B: simple tree graphs based on [24]. C: graph A with double depth and stochastic transitions. D, E, and F represent community graphs from [36], [28], and [34] respectively. (bottom) An example prompt for graph A. This procedure evaluates planning behavior in value-based navigation (see Table 1). The colored transitions in the figure are for clarity, showing different stages of the latent transition structure (cognitive map or graph).

for GPT-3 [49] as both pre-date GPT-4 and raise data contamination issues. Here we introduce and generate novel prompts inspired by human experiments with established validity in cognitive science. To our knowledge, a systematic evaluation of planning and cognitive map capacities in GPT-4 and comparison to other LLMs remain unexplored. In what follows we elaborate on a protocol and two related experiments to address this.

## 2 Methods

**The CogEval protocol.** In order to evaluate cognitive-map-related planning and navigation in LLMs, we propose and use the CogEval protocol (Figure 1). Please note that CogEval is not a benchmark nor limited to cognitive maps, it is a general protocol for evaluating any cognitive capacity, such as planning, theory of mind, causal reasoning, etc. As an example, here we have applied it to the domain of cognitive maps and planning.

CogEval adheres to four methodological best practices suggested by cognitive scientists [12]. First, the *latent construct or ability*: here we evaluate cognitive maps, which are representations that capture a model of the task structure, and adaptive planning, which requires an internal representation of task structures (similar to model-based RL [42] or task-oriented model-free RL [16–19]). Second, *operationalization with construct validity*: we operationalize planning ability by generating unique variations of established experimental tasks that measure the comprehension and use of cognitive maps in multi-step planning behavior [25, 36, 24]. Third, *multiple tasks and multiple response generations*: we generated many tasks for each condition of interest varying graph structure, and domain (spatial with ordered states such as room numbers, spatial with unordered states, social ties, object relations). Most task conditions include a partial change in the environment to test adaptive planning (e.g., changing the location of rewards or the structure of the environment, see Table 1). Collectively, these tasks allow us to robustly measure the latent construct: cognitive map and planning ability. Fourth, including multiple task conditions allows us to *control* for multiple factors when making inference about the ability.

Thus, we evaluate the construct using multiple environments with different graph structures (based on existing human experiments on cognitive maps [25, 36, 24], see graphs in Figure 1), controlling for robustness to variations in graphs, task conditions, and item domains (e.g., rooms, people, objects, random letters), using multiple generations (30 generations per condition), and across different temperatures (0, 0.5, and 1).

**LLMs evaluated.** We compared the following LLMs: GPT-4-* [2], GPT-3.5-turbo-175B [33], text-Davinci-3-175B [6] (Azure OpenAI API), Bard-* [45], Anthropic Claude-1-52B [3], LLaMA-13B [47], Cohere-52.4B [10], Alpaca-7B [43] (nat.dev API), where * means the number of parameters is undisclosed.

**Experiments.** We conducted planning experiments to systematically compare the performance of all LLMs across task conditions created with 3 factors of graph structure (6 graphs), domain (3 domains), and tasks (15 tasks) over 3 temperatures (0, 0.5, 1)).

### 2.1 A cognitive science inspired evaluation of cognitive maps and planning capacity in LLMs

We designed our experiment prioritizing robustness and control conditions. Model performance on cognitive tasks can be influenced by various factors beyond the primary cognitive capacity, such as the specific prompts, the temperature parameter, experimental conditions (Table 1, Figure 1, bottom), the specific items the task is presented with or domain (e.g., spatial connections vs. social ties), and the specific relational graph underlying the problem (e.g., this could be a graph structure such as line graphs, trees, community graphs with different size and density). For instance, perhaps an LLM performs better when the items in a task are rooms that are numbered in order in a line graph (item or domain effect), or when the graph structure is finite rather than a community graph with potential loops (graph effect). Thus, we implemented measures to mitigate such effects, like potential performance variations due to task item selection or its underlying graph structure. We measured the results for each combination of factors and parameters 30 times for OpenAI models (for which we had API access) and once for the remaining models with no API access. We compared the results across 10 LLMs.

*Why vary temperature?* Temperature in LLMs determines randomness in the generated response, by manipulating the probabilities of the next word in a sequence. Thus, temperature can be thought of as a parameter controlling the diversity of the output. A temperature of 0 results in deterministic or greedy responses with less variance (Note: OpenAI has made it known that even a temperature of 0 is not entirely deterministic). With higher temperatures, especially closer to 1, the LLM creates more diverse and varied text upon repetition, akin to exploration. While a higher temperature may be helpful for tasks that require varied responses or creativity, it could go either way for planning: on the

one hand, precision in planning trajectories may seem more in line with a deterministic temperature, and on the other, a higher temperature leads to exploration, which may improve behavior by getting out of local minima. Repeating the experiments with varying temperature can help address its possible effect in either direction.

**Statistical analysis.** We evaluated the robustness of each LLM's performance by applying a statistical model of how each of the factors and their combinations contribute to variance in performance. Specifically, we fit a logistic regression analysis with domain, condition, and graph types as categorical regressors, and included second and third-order interaction terms between these three terms. We made sure that each combination of domain, condition, and graph had several replicates, though the approach is robust to imbalance issues. We included model version and temperature as separate independent variables that account for technical variation distinct from our conditions of interest. See supplement for full details on analysis and results.

### 2.1.1 Task prompts

Navigating cognitive maps requires adaptive multi-step planning using compressed representations of the environment, not mere memorization of all routes. Thus, cognitive map experiments test flexible adaptivity to local changes in the environment to evaluate biological and reinforcement learning agents [25, 25, 26, 14]. Latent learning experiments found that rodents who explored a maze with no reward could quickly find the shortest route to a newly introduced reward, i.e., find an optimal policy in RL context. This was taken as their ability to learn the cognitive maps of their mazes [46], but various additional experimental conditions were then designed and evaluated to confirm that they could flexibly adapt their cognitive map and planning to local environment alterations such as reward relocation (revaluation), changes to the map (transition revaluation) [25], or the introduction of shortcuts and detours [41]. Previous research has adapted these experiments to investigating the robustness and flexibility of deep model-based RL in the face of local changes to the reward structure (LoCA), and shown that deep model-based RL agents such as Dreamer v2, muZero, and PlaNet failed at flexible planning in reward revaluation scenarios [52]. Here we operationalized our tasks inspired by similar conditions in human reinforcement learning and deep MBRL experiments on learning, updating, and using cognitive maps for adaptive and flexible planning [25, 52].

Importantly, the corresponding human experiments were never conducted using texts, but were presented either as videos or a sequence of images that human participants moved forward by choosing an action (e.g. pressing left, right, up, or down). We believe this mitigates the risks of contamination. Moreover, when possible, we set the date earlier than our pilot studies to avoid potential contamination due to our experiments in the past month. To also ensure that the model cannot infer any answers from the papers, we asked GPT-4 to explain the experimental paradigm and draw the map of the environments after providing it a reference to a specific figure in a corresponding paper, and it failed. Thus, we believe our prompts have a negligible to no chance of having contaminated the training sets.

Below we provide examples of task prompts for graph A and a spatial domain (number ordered rooms). All prompts are available in the supplementary material and on https://github.com/cogeval/cogmaps.

**I. Value-based or goal-driven planning.** Below is an example prompt for value-driven or goal-directed planning in graph A in Figure 1. Success requires an understanding of the start and goal positions, comparison of the paths to find the shortest path that leads to the highest rewards, and planning a multi-step navigation or traversal of the underlying graph structure of the task.

> *Imagine a world with six rooms. From the lobby you have two choices, room 1 and room 2. You enter room 1, at the end there's a door that leads to room 3, and room 3 leads to room 5. There's a chest in room 5. You open it and there's 10 dollars. Then you exit and start over. This time in the lobby you choose room 2, then enter room 4, which leads to room 6. There's a chest with 50 dollars. You return to the lobby. Which room will you choose to make the most money?*

**II. Transition Revaluation, after prompt I.** This condition occurs when the structure of the environment (e.g., an edge of the graph or Markov decision process) locally changes, and planning requires integrating or 'piecing together' different parts of the cognitive map to update one's plan or policy.

> *Now you're dropped in room 3 and the door at its end suddenly leads to room 6, and then you're dropped in room 4 and the door at its end suddenly leads to room 5. you return to the lobby. Which room will lead to more rewards?*

**III. Reward Revaluation, after prompt I.** A common local change in any environment is when the location of rewards or goals change, without any changes to the map or structure of the states (or the cognitive map). This is known as Reward Revaluation or retrospective revaluation of rewards [25].

> *Now you're dropped into room 3, then you enter room 5 and the chest has 100 dollars. Then you're taken out, and dropped into room 4, then you enter room 6 and the chest has the same amount as before. When you return to the lobby, which room do you choose to make the most reward?*

**V. Shortcut prompts with and without teleportation, after prompt I.** Tolman's experiments on cognitive maps [46] included a condition evaluating the animal's ability to discover shortcuts. Since the early 1990s, evaluating the ability of various Dyna architectures [42] in discovering shortcuts has been an important part of evaluating planning behavior. Below are two different shortcut prompts.

> *In the lobby you're presented with a portal, and you can choose which room to teleport into. Which room do you choose to maximize rewards?*

> *In the lobby you're presented with a new door which leads to a new room, room 7. Room 7's door leads directly to room 6. Remember that you will only be able to choose one path that leads to the most money. Which room from the lobby will lead to the path where one can make the most money?*

**V. Detour prompts with and without Teleportation, after prompt I.**

> *You enter the lobby and this time you encounter a new room, room 7. Room 7's door leads to room 8, and room 8 leads to room 9. From room 9 you can teleport anywhere. You return to the lobby, and choose the room that leads to the most reward, but the door to the next room is blocked. You go back to the lobby. Which room do you choose to reach the most rewards?*

> *You enter the lobby and this time you encounter a new room, room 7. Room 7's door leads to room 8, and room 8 leads to room 6. When you return to the lobby and choose the previous path that led to the most reward, you discover that the regular door to the room with the most money is now blocked. You go back to the lobby. You will only be able to choose one path that leads to the most money. Which room from the lobby will lead to the path where one can make the most money?*

## 3 Results

### 3.1 Repeated measures comparison of planning across LLMs

We evaluated out-of-the-box emergent or native ability of different LLMs on the cognitive map tasks. Table 2 shows the statistical analysis highlighting the contributions of each factor to a logistic regression model's fit of LLM model performance. The magnitude of chi-square test statistics indicate contribution to overall model fit. Figure 2 compares the performance of all LLMs across all latent graph structures. Table 3 shows mean and standard error for planning performance across tasks and LLMs.

The results in Table 2 indicate that the LLM ($\chi^2(11) = 2357.87$, p < .001), graph ($\chi^2(11) = 3431.53$, p < .001), condition ($\chi^2(11) = 2080.04$, p < .001), and domain ($\chi^2(11) = 304.06$, p < .001) each yielded significant chi-squared statistics. This means that not only did different LLMs performed

Table 1: Brief descriptions of the task conditions applied to varying graphs and domains

| Condition | Description | Group |
|---|---|---|
| **valuePath** | The optimal solution is to find the optimal policy, or shortest path, which yields the highest reward | |
| **1stepPath** | The optimal solution is a 1-hop policy, i.e., goal is adjacent to the starting state | |
| **2stepPath** | The optimal solution is a 2-step policy | **Traversal** |
| **3stepPath** | The optimal solution is a 3-step policy | |
| **nstepPath** | The optimal solution is an n-step policy, where max n is the diameter of the graph (longest shortest path) | |
| **rewardReval** | Upon a local change in the *reward structure*, the goal has changed and the optimal solution requires finding a new path | |
| **policyReval** | Upon a local change in the *reward structure*, the optimal solution requires finding a new policy | **RewReval** |
| **transReval** | Upon a local change in the *transition structure*, the goal is the same but the optimal solution requires finding a new policy | |
| **transRevalStochastic** | Upon a local change in the *transition structure*, the goal is the same but the optimal solution requires finding a new policy in a stochastic environment | **TransReval** |
| **nonteleShortcut** | Upon a change in the graph structure, the optimal solution requires finding a shortcut | |
| **nonteleShortcutCoT** | Upon a change in the graph structure, the optimal solution requires finding a shortcut, an additional CoT prompt is given | |
| **teleShortcut** | Upon a local change in the *transition structure*, the optimal solution requires finding a shortcut using a teleportation portal | **Shortcut** |
| **teleShortcutCoT** | Upon a local change in the *graph or transition structure*, the optimal solution requires finding a shortcut using a teleportation portal, an additional CoT prompt is given | |
| **nonteleDetour** | Upon a change in the graph structure, the optimal solution requires finding a detour | |
| **teleDetour** | Upon a local change in the *transition structure*, the optimal solution requires finding a detour using a teleportation step | **Detour** |

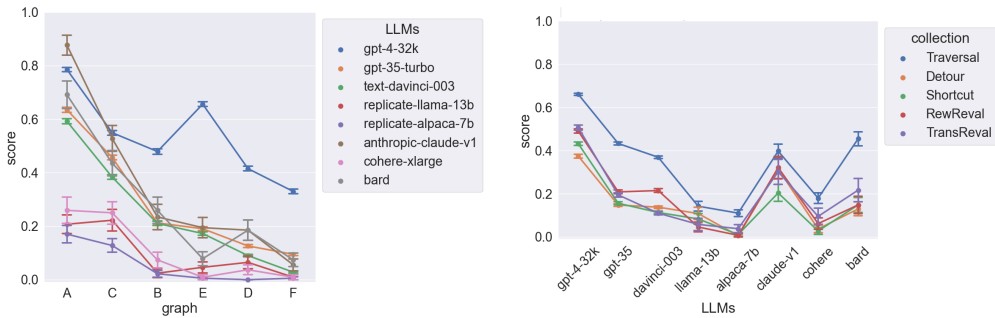

Figure 2: **Results for planning experiments in 8 LLMs.** (left) Mean and standard error of performance on all tasks for each of the different graphs (see Figure 1 for graph details) across different LLMs studied in this work. (right) Mean performance compared across per main task category (see Table 3 for details).

differently, but performance varied as a result of varying graphs, domains, and conditions. Conversely, the temperature showed a non-significant chi-squared statistic ($\chi^2(11) = 1.28, p = .53$) and the interaction between the LLM and temperature was also non-significant ($\chi^2(11) = 10.69, p = .71$). Noteworthy, the interactions among graph-domain, graph-condition, domain-condition, and graph-domain-condition were all significant (all p's < .001). The interactions among graph-domain ($\chi^2(11) = 334.41, p < .001$), graph-condition ($\chi^2(50) = 1651.33, p < .001$), domain-condition ($\chi^2(39) = 310.53, p < .001$), and graph-domain-condition ($\chi^2(108) = 1133.16, p < .001$) were all significant. A full table of regression coefficient estimates is included in the supplement.

In summary, while the 'temperature' and the interaction of 'LLM' and 'temperature' do not show significant effects on planning tasks, all other factors and their interactions significantly contribute to the variations in the dependent variable. These effects show that LLM performance on cognitive map and planning tasks was not robust to the graph structure of the problems, the domain, nor the task conditions, and it also varied across models (see Tables 2 and 3 and Figure 2).

### 3.2 Failure modes

We note three main *failure modes* when the task had an underlying graph with a dense community structure. Notably, when we probe the LLMs to list connected rooms or items as $(\text{state}, \text{actions}, \text{state})$ tuples, they do well (e.g., $(\text{room1}, \text{opendoor}, \text{room3})$ is a tuple for graph A in Figure 1). However, when asked to do any tasks with a community graph structure using this tuple knowledge, LLMs display the following failure modes; (1) hallucinate edges that do not exist, or (2) produce longer trajectories instead of shortest paths, or (3) produce trajectories that fall in loops. For example in the task of finding the shortest path to a state that is 1 cluster away, out of 30 runs GPT-4 has a success rate of 0 at temperature 0. Even with changing the temperature to 0.5 or

Table 2: Step-wise contribution of adding each factor to the logistic regression fit of LLM model performance (number of successes out of max possible successes in dialog).

| | term | Chi-squared Stat (Deviance) | df | p value |
|---|---|---|---|---|
| 1 | LLM | 2357.87 | 7 | <0.001 |
| 2 | graph | 3431.53 | 5 | <0.001 |
| 3 | domain | 458.74 | 2 | <0.001 |
| 4 | temperature | 1.28 | 2 | 0.53 |
| 5 | condition | 2080.04 | 4 | <0.001 |
| 6 | LLM and temperature | 10.69 | 14 | 0.71 |
| 7 | graph and domain | 334.41 | 10 | <0.001 |
| 8 | graph and condition | 1651.33 | 20 | <0.001 |
| 9 | domain and condition | 310.53 | 8 | <0.001 |
| 10 | graph, domain, condition | 1133.16 | 44 | <0.001 |

Table 3: Mean and standard errors for planning performance across all task conditions in all 10 LLMs. ARI scores closer to zero represent poor performance by the LLM and ARI scores reaching 1.0 represent performance matching Leiden.

| Condition | gpt-4-32k | gpt-35 | davinci-003 | claude-v1 | pythia-20b | cohere | llama-13b | alpaca-7b | bard |
|---|---|---|---|---|---|---|---|---|---|
| 1stepPath | **0.99, 0.08** | 0.76, 0.32 | 0.52, 0.45 | 0.57, 0.37 | 0.64, 0.41 | 0.27, 0.42 | 0.23, 0.38 | 0.27, 0.41 | 0.05, 0.10 |
| 2stepPath | **0.82, 0.35** | 0.73, 0.38 | 0.16, 0.25 | 0.61, 0.41 | 0.67, 0.42 | 0.29, 0.44 | 0.22, 0.37 | 0.35, 0.47 | 0.25, 0.50 |
| 3stepPath | 0.55, 0.38 | 0.37, 0.37 | **0.58, 0.43** | 0.27, 0.31 | 0.35, 0.49 | 0.04, 0.11 | 0.04, 0.07 | 0.06, 0.20 | 0.11, 0.10 |
| nonteleDetour | **0.55, 0.39** | 0.51, 0.35 | 0.55, 0.43 | 0.50, 0.41 | 0.51, 0.37 | 0.21, 0.35 | 0.19, 0.33 | 0.26, 0.38 | 0.29, 0.48 |
| nonteleShortcut | 0.56, 0.40 | 0.52, 0.39 | 0.49, 0.40 | **0.62, 0.43** | 0.40, 0.36 | 0.16, 0.27 | 0.11, 0.18 | 0.20, 0.30 | 0.29, 0.48 |
| nonteleShortcutCoT | **1.00, 0.00** | **1.00, 0.00** | 0.09, 0.07 | 0.58, 0.38 | 0.36, 0.38 | 0.37, 0.49 | 0.17, 0.29 | 0.37, 0.37 | - |
| nstepPath | **0.47, 0.38** | 0.31, 0.34 | 0.17, 0.27 | 0.33, 0.37 | 0.27, 0.42 | 0.05, 0.11 | 0.06, 0.08 | 0.12, 0.32 | 0.00, 0.00 |
| policyReval | 0.21, 0.18 | 0.18, 0.23 | 0.13, 0.04 | **0.28, 0.30** | 0.00, 0.00 | 0.00, 0.00 | 0.04, 0.07 | 0.05, 0.22 | 0.00, 0.00 |
| rewardReval | **0.67, 0.40** | 0.57, 0.36 | 0.34, 0.25 | 0.48, 0.35 | 0.60, 0.45 | 0.31, 0.44 | 0.28, 0.43 | 0.33, 0.44 | 0.14, 0.14 |
| teleDetour | 0.47, 0.35 | 0.34, 0.30 | **0.53, 0.44** | 0.37, 0.33 | 0.44, 0.41 | 0.21, 0.35 | 0.23, 0.37 | 0.23, 0.38 | 0.29, 0.48 |
| teleShortcut | **0.54, 0.39** | 0.35, 0.33 | 0.44, 0.41 | 0.45, 0.39 | 0.27, 0.33 | 0.16, 0.27 | 0.16, 0.22 | 0.12, 0.24 | 0.29, 0.48 |
| teleShortcutCoT | 0.50, 0.00 | 0.50, 0.00 | 0.04, 0.01 | 0.50, 0.50 | 0.39, 0.36 | 0.19, 0.40 | **0.83, 0.29** | 0.35, 0.36 | - |
| transReval | **0.60, 0.42** | 0.59, 0.40 | 0.49, 0.38 | 0.55, 0.36 | 0.47, 0.42 | 0.19, 0.28 | 0.22, 0.33 | 0.27, 0.37 | 0.08, 0.17 |
| transRevalStochastic | 0.73, 0.36 | 0.52, 0.36 | **0.91, 0.24** | 0.78, 0.34 | 0.36, 0.32 | 0.00, 0.00 | 0.11, 0.19 | 0.22, 0.39 | - |
| valuePath | 0.58, 0.41 | 0.66, 0.40 | **0.66, 0.39** | 0.44, 0.41 | 0.49, 0.46 | 0.31, 0.40 | 0.27, 0.39 | 0.33, 0.45 | 0.29, 0.48 |

1 and repeating the same 30 runs its success rate can not exceed 10%. Please refer to Figure 3 for examples of above failure modes.

## 4 Discussion and future directions

This paper makes two main contributions. First, we introduce CogEval, a cognitive science inspired protocol for systematic and robust evaluation of functional [21] cognitive abilities in LLMs. Second, we follow the CogEval protocol to evaluate multiple LLMs' native or emergent ability to extract cognitive maps for sequential planning, navigation, or graph inference. All tasks and prompts are based on non-linguistic human cognitive science experiments that we adapted into text prompts for the first time. We test for robustness of the findings by varying task conditions, graph structure, domains (spatial, social), and LLM temperature. Our systematic evaluation reveals that while LLMs display apparent competence on some tasks in simpler graphs, they do not have out-of-the-box zero-shot emergent cognitive map comprehension or planning competence for other graphs.

*Methodological contribution.* **CogEval**, our proposed cognitive-science inspired protocol [12] for systematic evaluation of LLMs, affords the following contributions. (1) We avoid the reuse of contaminated standardized benchmarks by creating novel prompts based on non-text-based experiments that are known to evaluate cognitive maps and planning in humans, animals, and RL. (2) We use multiple tasks to probe the cognitive constructs (cognitive maps and planning) and repeat each interaction multiple times and across different temperatures. (3) We use statistical analysis to evaluate the robustness and reliability of each effect, with three main factors of graph structure, item domain (spatial vs. social), and task condition (e.g., value-based decision making, shortcut, detour, see Table 1). (4) We employ chain of thought and instruction prompts to evaluate the limits of out-of-the-box cognitive abilities of LLMs (Supplementary Experiment 2), and (5) analyze and categorize different types of failure modes. Please note that CogEval is not a benchmark nor limited to evaluating cognitive maps and planning, it is a general protocol for evaluating any cognitive capacity in LLMs. As an example, in this paper we have applied it to the domain of cognitive maps and planning.

*No evidence for understanding cognitive maps or planning.* Our systematic and incremental evaluations reveal limited to no cognitive map capacities in the current generation of LLMs - including

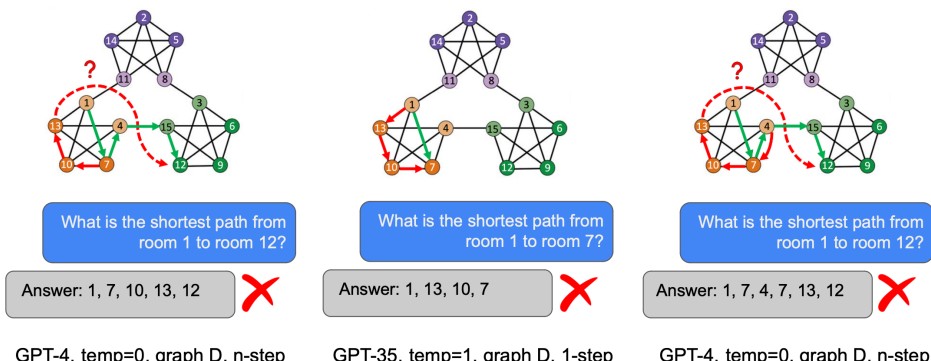

Figure 3: **Examples of three failure modes.** (left) Edge hallucination. (middle) Failure at finding a 1-step policy within the same cluster. (right) Failure at multi-hop path by both falling in a loop and hallucinating edges. In each example the blue box is the *task prompt*, the grey box shows the *model response*, and the green arrows demonstrate the correct response on the graph.

GPT-4. Specifically, we find that LLMs only show apparent competence on simple sequential inference tasks where route memorization can help, and given LLMs have received all possible trajectories in the text prompt. We also observe that the sparsity of graph structure drove performance. However, when 1-step and multi-step traversal and planning require understanding the underlying relational structure of the environment graph, LLMs including GPT-4 fail by hallucinations, suboptimally long routes, or falling in loops.

*How did LLMs solve the simpler tasks?* Without access to full architectures or training sets, we can only form hypotheses based on our behavioral observations. We observe that LLMs do better in problems where the entire trajectories are explicitly available in the text prompts, and they only need to retrieve and piece together partial changes. However, planning behavior in more complex graphs is far worse, and this is not just due to graph size: performance was worse for graph B (7-node tree) than C (12 node, parallel lines). Performance on the 15-node graph with 3 dense clusters was worse than the 16-node (4-cluster) graph that has better cross-cluster connectivity and more paths among clusters.

These observations suggest that LLMs may fail at planning problems where they need to use the transition structure to unroll the trajectories and find the correct path, which is closer to the notion of planning in model-based RL and in cognitive science. Capturing the underlying structure and using it to unroll trajectories are quintessential to cognitive maps and planning ability. Thus, the apparent competence in simpler tasks may be due to using cached or memorized routes rather than understanding the cognitive map, planning, or inference ability.

LLMs may do better in smaller and simpler graphs because the prompt already expands all the possible paths or trajectories. When there is a change in the rewards or transition structure, LLMs only need to change one step in an already laid out path. However, in more complex graphs only the one-step connections are laid out in the prompt, but not all paths or trajectories between any given two nodes. We observed that failures significantly increase in tasks with these larger graphs with community structures, even when an LLM can list the pairwise tuples of connected states (see failure modes, Figure 3).

*Interpreting the results.* The experiments in the paper are not meant to be interpreted as a benchmark for planning. They probe the same construct in different ways, evaluating the ability to use information about $(\text{state}, \text{action}, \text{state})$ tuples, e.g., $(\text{room1}, \text{openleftdoor}, \text{room3})$, to piece together policies in response to task prompts. A reader may wonder why we claim that LLMs do not display emergent planning in spite of non-zero performance for some tasks in our experiments (Figure 2. We interpret the findings as such due to failure modes and various inconsistencies in success cases (Figure 3). For instance, a common failure mode is generating sequences with hallucinated $(\text{state}, \text{actions}, \text{state})$ tuples that do not exist. For GPT-4 this constitutes over 20% of 4-step plans. Another common failure mode is that they fall into loops when prompted to find the shortest path between two states (Figure 3, left and right). LLMs sometimes even fail to identify 1-step paths or suggest multi-hop trajectories for traversing to an adjacent state (Figure 3, middle).

These observations seem counter-intuitive given some LLMs can generate a list of tuples when asked, but fail to use the tuples to make (even 1-step) valid plans. This shows that, while LLMs appear to solve planning problems when given simple routes that can be explicitly memorized, they cannot *generalize* from route memory solutions (trajectories directly in the prompt) to using the tuples to adaptively generate branching plans. Together, these inconsistent observations are in line with the hypothesis that *LLMs do not understand cognitive maps* and therefore cannot consistently plan. Elsewhere propose black box architectures that improve planning performance [54]. However, our findings point to a lack of out-of-the-box zero-shot emergent planning ability.

*Limitations.* First, we lack knowledge of LLMs like GPT-4's architecture or training. To address this limitation, we did not use existing text-based benchmarks and instead generated novel prompts not in their training sets. Second, in the human experiments that influenced our prompts participants learn gradually, experiencing states one-by-one, only tested after they showed signs of learning, similar to a model-based RL agent having the transition structure and using it for inference and planning. To address this difference, we present the environment's structure in linguistic format. The LLM had to extract the cognitive map, identify the goal location based on instructions. and infer the policy towards the goal. Third, we do not have a human baseline for the language-based tasks, so a comparison with human behavior on the exact language prompts remains the topic of future studies.

*Implication for applications.* LLMs are expected to be applied in fields like gaming, planning, and social reasoning, with tasks that require understanding the inherent relational structure of the problem from the input for flexible reasoning and planning. However, here we show various failure modes in the understanding of the underlying cognitive maps or planning abilities, including hallucination and falling in loops. Even when provided instructions and Chain of Thought (CoT) prompts like breadth-first search (BFS), we observe that GPT-4 struggles to process multi-hop paths it has not experienced (Supplementary Experiment 2). These findings suggest caution in the application of LLMs in problems that involve planning or complex structures. Below we discuss future directions that may mitigate these challenges for problems with simpler structures.

*LLMs as programmable machines rather than emergent intelligence?* While some regard LLMs as agents with emergent intelligence comparable to humans and animals, our findings are more consistent with the view that LLMs are programmable machines where natural language is their programming language [20]. Thus, here we evaluated planning in LLMs in a functionalist and multiple-realizability sense rather than making any assumptions of them being "human-like" [27].

*Future directions.* A future direction is to analyze embedding representations and attention in LLMs, and test hypotheses about representations underlying success and failure in planning. This mirrors how neuroscience analyzes neural data to understand representations in model-based and predictive planning and decision-making [25, 7]. A recent study suggests a promising future for this direction [57]. Another interesting direction is to study the limits of LLMs' transitive inference using pairwise associations [37, 34], given we observed that while some LLMs could list pairwise tuples or recognize the goal, they still struggled with planning. A further direction is to study whether the use of *schemas*, i.e., overused, generalized cognitive maps such as "airport" [50, 22, 11, 51], can improve performance on real-world scenarios by evoking helpful structures, given LLMs have shown promise with analogical reasoning tasks [53]. Finally, some have suggested ways to improve planning by augmenting LLMs with algorithms that enable executive control, an interesting direction that can contribute to the future of augmenting both larger and especially smaller language models (see [54]).

*LLMs need a hippocampus and prefrontal cortex.* The hippocampus and prefrontal cortex in the brain extract the relational structure or cognitive maps from sequential data to flexibly plan at multiple scales [7, 26]. Their functions can inspire memory, planning, and executive control augmentations to LLMs in order to mitigate failure modes such as hallucinating edges. Ongoing research shows that indeed prefrontal cortex-inspired solutions can improve planning performance in an LLM-based architecture with multiple calls to GPT-4 [54], pointing at a promising future direction.

*Summary.* We introduced CogEval, a cognitive science inspired protocol for systematic and robust evaluation of LLMs. We applied CogEval to evaluate planning performance across 8 LLMs and found poor performance.

## 5  Acknowledgement

We are extremely grateful to Peter Lee, Michael Frank, John Krakauer, Joshua Tenenbaum, Paul Bennett, Alison Gopnik, and Melanie Mitchell for early feedback on methods and results. We would also like to acknowledge Derek Worthen and Richard Ciapala for engineering support.

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
