# Evaluating Cognitive Maps and Planning in Large Language Models with CogEval (Supplementary Materials)

**Ida Momennejad**[*]
Microsoft Research
New York, NY
`idamo`

**Hosein Hasanbeig**[*]
Microsoft Research
New York, NY
`hosein.hasanbeig`

**Felipe Vieira Frujeri**[*]
Microsoft
Redmond, WA
`felipe.frujeri`

**Hiteshi Sharma**
Microsoft
Redmond, WA
`hiteshi.sharma`

**Robert Ness**
Microsoft Research
Redmond, WA
`robertness`

**Nebojsa Jojic**
Microsoft Research
Redmond, WA
`jojic`

**Hamid Palangi**
Microsoft Research
Redmond, WA
`hpalangi`

**Jonathan Larson**
Microsoft Research
Redmond, WA
`jolarso`

`@microsoft.com`

## 1  Supplementary Experiment 1: Systematic graph explorations

To systematically evaluate GPT-4's planning or graph traversal failure modes, we created a three-block community graph structures where each block contains five vertices. Using this approach, we vary the connection density within each community block and ask GPT-4 to perform reasoning tasks over each permutation of the graph structure as block density is varied. For the graph community block model, example graphs are shown in Figure 1 with the community graphs starting as simple line graphs on the left - representing the sparsest level of connectivity. We then create a new edge within each block for each iteration of the experiment until each community block forms a clique structure as seen on the right of Figure 1. To measure performance, the LLM is asked to assign partitions for each vertex such as to maximize each graph's modularity. Modularity is chosen as the task as it requires a *non-trivial understanding of the graph* beyond the local network of any single vertex in order to detect the boundaries between communities. The LLM's vertex assignment is then compared to the vertex assignments obtained from a Leiden [11] modularity maximization process. The results are compared using Adjusted Rand Index (ARI), which gives a similarity score between the actual modularity-maximized partitioning scheme and the observed partitioning that the LLM returned. ARI scores closer to zero represent poor performance by the LLM and ARI scores reaching 1.0 represent performance matching Leiden. This is performed for temperatures 0.05, 0.5, 0.95 and TopP 0.05, 0.5, 0.95. Each configured test is executed thirty times through the GPT-4 chat completion API.

---

[*]Equal contribution

37th Conference on Neural Information Processing Systems (NeurIPS 2023).

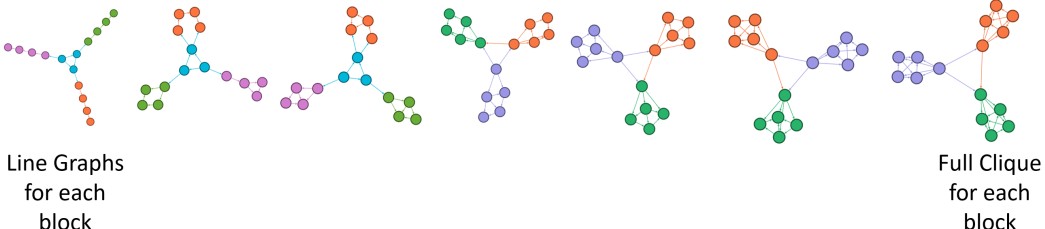

**Systematic exploration of three interconnected community blocks with varying density**

Line Graphs for each block

Full Clique for each block

Figure 1: **Systematic investigation of line graphs to full-clique structures.** Each graph is composed of three blocks, where each block is its own sub-graph structure with five interconnected vertices on which we vary the edge density. Each block has a bridge node linked to other blocks' bridge nodes. The left graph displays the sparsest density with a line graph for each block, while the right graph shows the densest blocks with each forming a fully interconnected clique. Vertex partition assignments are color-coded via Leiden modularity maximization.

## 1.1 Supplementary Experiment 1 Evaluation: Evaluating the systematic effect of graph structure

Figure 7 shows the result of systematic evaluation over the community based graph structures using GPT-4. The y-axis measures how well the LLM performed (higher is better) and the x-axis measures the density of each of the community block structure. Outlier points are added directly to each plot and each panel shows how performance varies with temperature. These findings suggest that the LLM performs more poorly in the sparse community structures, but better as the edge density increases. Additionally, a higher temperature results in poorer performance with higher variance. These results may appear in contrast to the results shown in Experiment 1, where LLMs show apparent success in *local traversal* of a small graph. However, as the task changed from local traversal (Experiment 1) towards optimization and reasoning of a non-local graph structure (Supplementary Experiment), we observe the LLMs fail when the structures are sparse. This points to the LLM's struggle to reason over local neighborhood and community structures, and is consistent with failure on community graphs as demonstrated in Experiment 1. Moreover, these findings show how *the LLMs' behavior is not consistent* over community structures vs. simple traversals, varying across overall sparsity. This is in line with the hypothesis that LLMs lack functional understanding of underlying structures of the problems, which may contribute to their failure in multi-step planning.

In order to better understand failure modes, this experiment systematically evaluates the observed poor performance of GPT-4 on community graphs on gradually more dense community graphs.

## 1.2 Supplementary Experiment 1: Evaluating the systematic effect of graph structure as TopP is varied

Below are the graphs for each TopP configuration:Systematic graph explorations and the variance of TopP

## 1.3 Supplementary Experiment 1: Prompt templates

Figure 6 shows the prompt templates that were used in this supplementary experiment.

## 2 Supplementary Experiment 2: Evaluating the effect of Chain of Though (CoT) instructed prompts

LLM evaluation is usually performed within the in-context learning framework [1, 8], where the input to the LLM is the text of the problem to be solved, preceded with several examples of related problems and their solutions, possibly worked out step-by-step. The rationale is that a single problem may be ambiguously stated (as far as the LLM is concerned), and a few examples, possibly with explanations, may be all that is needed to disambiguate the intent. However, the choice and even the

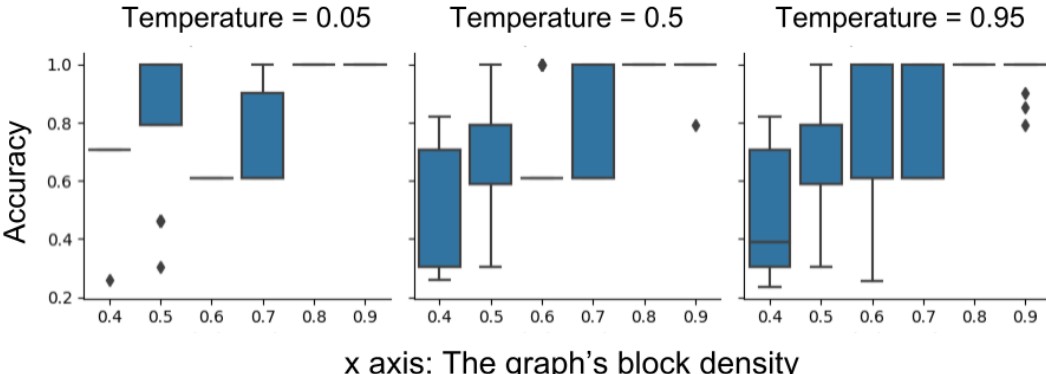

Figure 2: **Supplementary Experiment 1 results.** Systematic investigation of line graphs to full-clique structures. y-axis: accuracy of the LLM as measured using the Adjusted Rand Index (ARI) between a maximal modularity partitioning from the LLM as compared to observed maximal modularity via Leiden. An ARI of 1.0 indicates the LLM matched Leiden. x-axis: the density of each block, 0.4 represents a line graph of the five vertices and 1.0 represents a fully connected clique structure. Temperature: 0.05, 0.5, 0.95,TopP: 0.95.

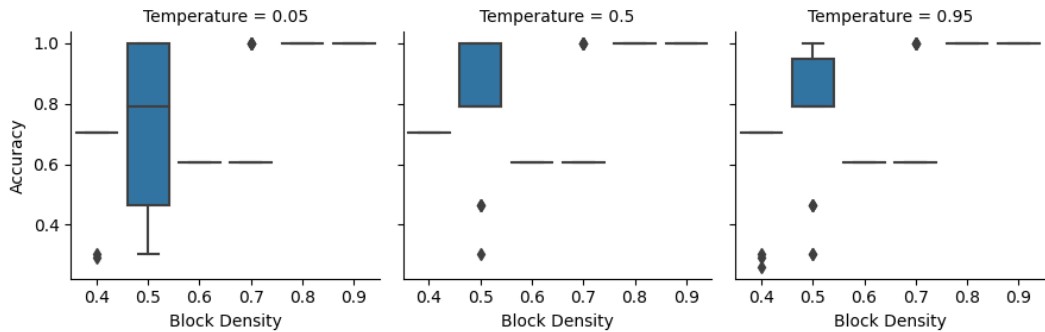

Figure 3: Results of systematic investigation of line graphs to full-clique structures. TopP = 0.05.

order of examples impacts the performance [7], as does the incorporation of auxiliary knowledge, [10, 15, 9], particularly in the form of Chain-of-Thought (CoT) reasoning [14, 13, 16, 2, 12, 6, 4, 5].

While CoT prompting is not a rigorously defined concept, a prompt with a small number of worked out examples, serving as an instance of few-shot learning, may qualify as a CoT prompt and has been shown to improve performance considerably on cognitive tasks (e.g., Theory of Mind [8]). However, such a prompt can be so regimented that they effectively turn an LLM into a Turing Machine executing a given algorithm the way a computer would [3].In this view, careful CoT prompts could have a significant effect both on the performance and on our interpretation of how it is achieved.

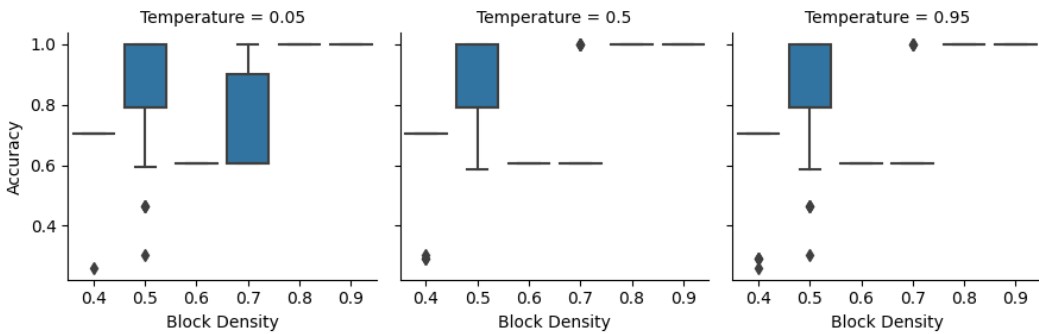

Figure 4: Results of systematic investigation of line graphs to full-clique structures. TopP = 0.5.

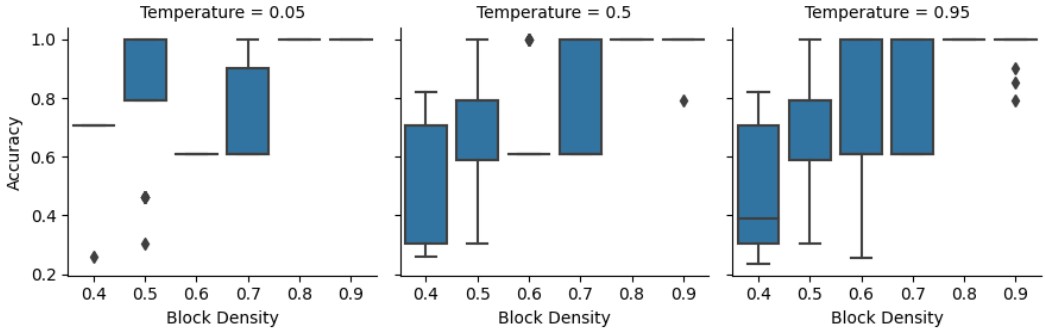

Figure 5: Results of systematic investigation of line graphs to full-clique structures. TopP = 0.95.

```
prompt_id,calculation,prompt_template,prompt,token_count,response
prompt_01,modularity,"Given the following comma delimited edge list of source and target nodes in a graph, assign each node a cluster as to maximize the
modularity of the graph. Provide the results in JSON format with the key representing the node and the value representing the community.

Example:
Source,Target
1,2
2,1
3,4
4,3

Answer:
{
    ""1"": 1,
    ""2"": 1,
    ""3"": 2,
    ""4"": 2
}

Graph to Analyze
=========
Source,Target
{EDGE_LIST}
=========",,,
```

Figure 6: Prompt template for maximizing modularity

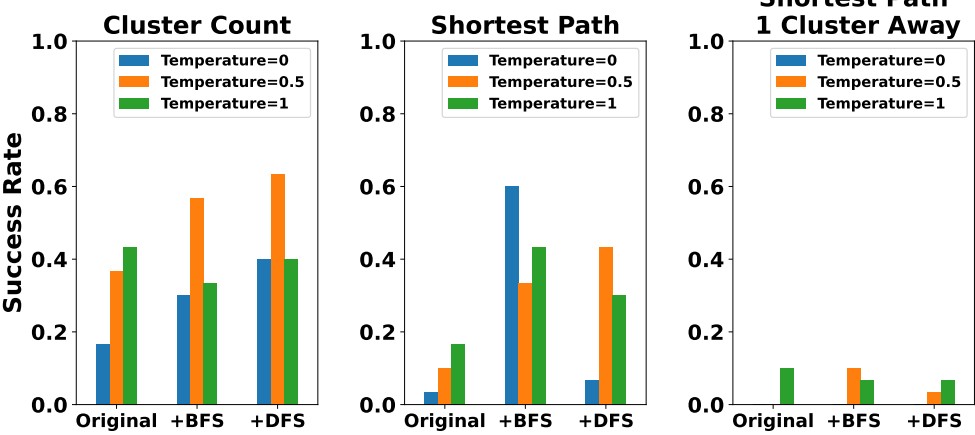

Figure 7: **Experiment 2 results.** (Bottom) BFS and DFS instructions marginally enhance performance on community graphs. In the Cluster counting task (graph D) adding BFS or DFS is beneficial at temperatures 0 and 0.5 but less at 1. For finding shortest paths within a cluster, BFS or DFS help with BFS being effective at temperature 0. However, for finding the shortest path 1-cluster away, only BFS at temperature 0.5 yields slight improvements.

Here wetried breadth first and depth first instructions as follows:

## 2.1 BFS (Breadth First Search) instruction:

"Think carefully before you respond. You can try using Breadth-first search (BFS), it is a graph traversal algorithm that visits all the vertices of a graph in breadth-first order, starting from a given source vertex. In BFS, vertices are visited in layers, where the vertices at distance 1 from the source vertex are visited first, followed by the vertices at distance 2, and so on. BFS uses a queue data structure to keep track of the vertices to be visited, and it ensures that no vertex is visited more than once. BFS is useful for finding the shortest path between two vertices in an unweighted graph, or for exploring all the vertices in a graph."

## 2.2 DFS (Depth First Search) instruction:

"Think carefully before you respond. You can try using Depth-first search (DFS), it is a graph traversal algorithm that visits all the vertices of a graph in depth-first order, starting from a given source vertex. In DFS, the algorithm traverses as far as possible along each branch before backtracking. DFS uses a stack data structure to keep track of the vertices to be visited, and it ensures that all vertices connected to a visited vertex are explored before backtracking. DFS is useful for finding cycles in a graph, for exploring all the vertices in a graph, or for finding a path between two vertices. However, unlike BFS, DFS does not guarantee that the shortest path is found."

We explored how the simple instructions impact LLM performance for different temperatures to investigate if the effectiveness of a given prompt can be impacted by the level of uncertainty caused by the temperature parameter. We find that while in some cases the performance improves, the effects are not consistent nor monotonic. This is an interesting phenomenon that needs further investigation to be better understood.

## 2.3 Supplementary Experiment 2: CoT prompts used for BFS and DFS

We have used quite general and simple description of graph traversal algorithms in the prompt to measure how much the LLM can leverage the information in the prompt. The prompts that we have used for BFS and DFS are as follows and they are directly appended to the existing prompt for each task.

- **BFS**: "*Think carefully before you respond. You can try using Breadth-first search (BFS), it is a graph traversal algorithm that visits all the vertices of a graph in breadth-first order, starting from a given source vertex. In BFS, vertices are visited in layers, where the vertices at distance 1 from the source vertex are visited first, followed by the vertices at distance 2, and so on. BFS uses a queue data structure to keep track of the vertices to be visited, and it ensures that no vertex is visited more than once. BFS is useful for finding the shortest path between two vertices in an unweighted graph, or for exploring all the vertices in a graph.\endofprompt\*"

- **DFS**: "*Think carefully before you respond. You can try using Depth-first search (DFS), it is a graph traversal algorithm that visits all the vertices of a graph in depth-first order, starting from a given source vertex. In DFS, the algorithm traverses as far as possible along each branch before backtracking. DFS uses a stack data structure to keep track of the vertices to be visited, and it ensures that all vertices connected to a visited vertex are explored before backtracking. DFS is useful for finding cycles in a graph, for exploring all the vertices in a graph, or for finding a path between two vertices. However, unlike BFS, DFS does not guarantee that the shortest path is found.\endofprompt\*"

# 3 Brief descriptions of the task conditions applied to varying graphs and domains

Table 1: Brief descriptions of the task conditions applied to varying graphs and domains

| Condition | Description | Group |
|---|---|---|
| **valuePath** | The optimal solution is to find the optimal policy, or shortest path, which yields the highest reward | |
| **1stepPath** | The optimal solution is a 1-hop policy, i.e., goal is adjacent to the starting state | |
| **2stepPath** | The optimal solution is a 2-step policy | **Traversal** |
| **3stepPath** | The optimal solution is a 3-step policy | |
| **nstepPath** | The optimal solution is an n-step policy, where max n is the diameter of the graph (longest shortest path) | |
| **rewardReval** | Upon a local change in the *reward structure*, the goal has changed and the optimal solution requires finding a new path | **RewReval** |
| **policyReval** | Upon a local change in the *reward structure*, the optimal solution requires finding a new policy | |
| **transReval** | Upon a local change in the *transition structure*, the goal is the same but the optimal solution requires finding a new policy | **TransReval** |
| **transRevalStochastic** | Upon a local change in the *transition structure*, the goal is the same but the optimal solution requires finding a new policy in a stochastic environment | |
| **nonteleShortcut** | Upon a change in the graph structure, the optimal solution requires finding a shortcut | |
| **nonteleShortcutCoT** | Upon a change in the graph structure, the optimal solution requires finding a shortcut, an additional CoT prompt is given | **Shortcut** |
| **teleShortcut** | Upon a local change in the *transition structure*, the optimal solution requires finding a shortcut using a teleportation portal | |
| **teleShortcutCoT** | Upon a local change in the *graph or transition structure*, the optimal solution requires finding a shortcut using a teleportation portal, an additional CoT prompt is given | |
| **nonteleDetour** | Upon a change in the graph structure, the optimal solution requires finding a detour | **Detour** |
| **teleDetour** | Upon a local change in the *transition structure*, the optimal solution requires finding a detour using a teleportation step | |

# 4 Summary of high-level statistical analysis

We chose a logistic regression to model the number of items the LLM answers correctly in a given dialog out of a total number of possible correct answers. We aggregated the results into an analysis of deviance table (the generalized linear model equivalent of Analysis of Variance or ANOVA), which highlights the contributions of each factor and their interactions to performance, along with significance statistics.

In the presented study, the "score" of a dialog is the number of correct answers provided by the LLM out of a total number of correct answers possible for that dialog. We modeled the score using a logistic regression approach; the score follows a binomial distribution with a probability parameter determined by the three categorical variables (graph structure, condition, and domain) as well as model and temperature. We our regression model included second and third-order interaction terms between levels of these three terms.

Our initial strategy was to assume that for a particular combination of the three factors (graph structure, condition, and domain), the conjunction of model and temperature could be likened to a 'subject' in a repeated measures analysis. With the inability to set a seed in these LLMs, we posited that each repeated measurement for the engine and temperature variables could be akin to a repeated replicate measure in a longitudinal or panel study, where there would be "within-subject variation" across replicates. We introduced a nested random effect (temperature nested within model) to the linear component of a linear regression model. Note that we did not have an equal number of replicates across each combination of graph structure, condition, domain, model, and temperature (the minimum number of replicates for a combination was 1, the maximum was 30, and the mean was 7.1). However, the logistic regression approach is robust to replicate imbalance.

We used a two-step fitting process. In the first step, we used elastic net to fit the model using the R package glmnet. We relied on elastic net's mix of L1 and L2 regularization to address cases of LLMs where we collected less data, and to address the multicollinearity introduced by the interaction terms. The parameter estimates are shown in Table 2. Next, we refit a new model using non-regularized logistic regression on the predictors with non-zero coefficient estimates in the first model, and used this model to generate the analysis of variance (deviance) table in Table **??**. Deviance is a measure of goodness of fit; it quantifies the discrepancy between the observed scores and the scores predicted by the model. Each row in the Chi-squared statistic column quantifies the reduction in deviance by adding the categorical variables associated with the term in that row, given the variables from the previous rows are included in the model.

Table 2: Parameter estimates from the regularized logistic regression model. Baselines are condition:traversal, graph:n7line, domain:ordRooms, LLM:replicate-alpaca-7b, temp:0. NA values indicate the data was not sufficient to fit the parameters.

| | factor | level | estimate | odds multiple | p value |
|---|---|---|---|---|---|
| 1 | (Intercept) | (Intercept) | 0.85 | 2.33 | <0.001 |
| 2 | LLM | bard | -0.41 | 0.66 | 0.01 |
| 3 | LLM | cohere-xlarge | -2.25 | 0.11 | <0.001 |
| 4 | LLM | gpt-35-turbo | -0.40 | 0.67 | <0.001 |
| 5 | LLM | gpt-4-32k | 1.25 | 3.50 | <0.001 |
| 6 | LLM | replicate-alpaca-7b | -3.57 | 0.03 | <0.001 |
| 7 | LLM | replicate-llama-13b | -2.67 | 0.07 | <0.001 |
| 8 | LLM | text-davinci-003 | -0.73 | 0.48 | <0.001 |
| 9 | graph | n13line | -1.14 | 0.32 | <0.001 |
| 10 | graph | n7tree | -2.59 | 0.07 | <0.001 |
| 11 | graph | n15star | -1.47 | 0.23 | <0.001 |
| 12 | graph | n21star | -1.78 | 0.17 | <0.001 |
| 13 | graph | n16cluster | -0.62 | 0.54 | <0.001 |
| 14 | domain | socialTies | 0.58 | 1.78 | <0.001 |
| 15 | domain | unordSpatial | -1.25 | 0.29 | <0.001 |
| 16 | temperature | 0.5 | -0.00 | 1.00 | 0.96 |
| 17 | temperature | 1 | -0.06 | 0.95 | 0.12 |
| 18 | condition | Detour | -2.35 | 0.10 | <0.001 |
| 19 | condition | RewReval | -2.56 | 0.08 | <0.001 |
| 20 | condition | Shortcut | -2.01 | 0.13 | <0.001 |
| 21 | condition | TransReval | -2.23 | 0.11 | <0.001 |
| 22 | model and temp | gpt-4-32k.0.5 | 0.04 | 1.04 | 0.52 |
| 23 | model and temp | gpt-4-32k.1 | 0.05 | 1.05 | 0.37 |
| 24 | graph and domain | n7line & ordRooms | 2.11 | 8.27 | <0.001 |
| 25 | graph and domain | n7tree & ordRooms | 1.01 | 2.75 | <0.001 |
| 26 | graph and domain | n7line & socialTies | 0.77 | 2.16 | <0.001 |
| 27 | graph and domain | n7line & unordSpatial | NA | NA | NA |
| 28 | graph and condition | n7line & Detour | 1.95 | 7.02 | <0.001 |
| 29 | graph and condition | n13line & RewReval | 3.18 | 23.99 | <0.001 |
| 30 | graph and condition | n16cluster & RewReval | 1.46 | 4.30 | <0.001 |
| 31 | graph and condition | n7line & Shortcut | 1.66 | 5.28 | <0.001 |
| 32 | graph and condition | n13line & Traversal | 2.06 | 7.88 | <0.001 |
| 33 | graph and condition | n16cluster & Traversal | 0.13 | 1.14 | 0.19 |
| 34 | graph and condition | n7line & Traversal | 1.32 | 3.76 | <0.001 |
| 35 | graph and condition | n7tree & Traversal | 1.97 | 7.17 | <0.001 |
| 36 | domain and condition | socialTies & Shortcut | -0.02 | 0.98 | 0.90 |
| 37 | domain and condition | ordRooms & Traversal | -0.48 | 0.62 | <0.001 |
| 38 | domain and condition | socialTies & Traversal | -0.99 | 0.37 | <0.001 |
| 39 | graph, domain, condition | n16cluster & ordRooms & Detour | 0.70 | 2.02 | <0.001 |
| 40 | graph, domain, condition | n7line & ordRooms & Detour | 0.50 | 1.65 | 0.10 |
| 41 | graph, domain, condition | n7tree & ordRooms & Detour | 2.39 | 10.94 | <0.001 |
| 42 | graph, domain, condition | n15star & socialTies & Detour | 1.30 | 3.69 | <0.001 |
| 43 | graph, domain, condition | n21star & socialTies & Detour | 1.01 | 2.73 | <0.001 |
| 44 | graph, domain, condition | n7line & socialTies & Detour | -1.06 | 0.35 | <0.001 |
| 45 | graph, domain, condition | n15star & unordSpatial & Detour | 1.78 | 5.92 | <0.001 |
| 46 | graph, domain, condition | n21star & unordSpatial & Detour | 2.24 | 9.40 | <0.001 |
| 47 | graph, domain, condition | n13line & ordRooms & RewReval | -0.57 | 0.57 | 0.01 |
| 48 | graph, domain, condition | n7tree & ordRooms & RewReval | 2.58 | 13.20 | <0.001 |
| 49 | graph, domain, condition | n13line & socialTies & RewReval | -0.20 | 0.82 | 0.38 |
| 50 | graph, domain, condition | n15star & socialTies & RewReval | 2.58 | 13.25 | <0.001 |
| 51 | graph, domain, condition | n16cluster & socialTies & RewReval | 0.80 | 2.23 | <0.001 |
| 52 | graph, domain, condition | n15star & unordSpatial & RewReval | 3.61 | 36.98 | <0.001 |
| 53 | graph, domain, condition | n16cluster & unordSpatial & RewReval | 2.07 | 7.94 | <0.001 |
| 54 | graph, domain, condition | n7line & unordSpatial & RewReval | 2.58 | 13.21 | <0.001 |
| 55 | graph, domain, condition | n7line & ordRooms & Shortcut | -1.76 | 0.17 | <0.001 |
| 56 | graph, domain, condition | n21star & socialTies & Shortcut | 0.91 | 2.48 | <0.001 |
| 57 | graph, domain, condition | n7line & socialTies & Shortcut | -0.48 | 0.62 | 0.06 |
| 58 | graph, domain, condition | n7tree & socialTies & Shortcut | 3.59 | 36.40 | <0.001 |
| 59 | graph, domain, condition | n15star & unordSpatial & Shortcut | 1.95 | 7.05 | <0.001 |
| 60 | graph, domain, condition | n16cluster & unordSpatial & Shortcut | 1.28 | 3.59 | <0.001 |
| 61 | graph, domain, condition | n13line & ordRooms & TransReval | 2.31 | 10.12 | <0.001 |
| 62 | graph, domain, condition | n15star & ordRooms & TransReval | 1.04 | 2.82 | <0.001 |
| 63 | graph, domain, condition | n7tree & ordRooms & TransReval | 2.84 | 17.13 | <0.001 |
| 64 | graph, domain, condition | n7tree & socialTies & TransReval | 3.20 | 24.49 | <0.001 |
| 65 | graph, domain, condition | n13line & unordSpatial & TransReval | 3.37 | 29.19 | <0.001 |
| 66 | graph, domain, condition | n16cluster & unordSpatial & TransReval | 1.78 | 5.92 | <0.001 |
| 67 | graph, domain, condition | n7line & unordSpatial & TransReval | 2.65 | 14.15 | <0.001 |
| 68 | graph, domain, condition | n13line & ordRooms & Traversal | 1.70 | 5.47 | <0.001 |
| 69 | graph, domain, condition | n7line & ordRooms & Traversal | -0.67 | 0.51 | 0.01 |
| 70 | graph, domain, condition | n13line & socialTies & Traversal | 0.77 | 2.15 | <0.001 |
| 71 | graph, domain, condition | n16cluster & socialTies & Traversal | -0.02 | 0.98 | 0.87 |
| 72 | graph, domain, condition | n7line & socialTies & Traversal | NA | NA | NA |
| 73 | graph, domain, condition | n21star & unordSpatial & Traversal | 0.65 | 1.91 | <0.001 |
| 74 | graph, domain, condition | n7tree & unordSpatial & Traversal | 0.72 | 2.06 | <0.001 |

# 5   Experiment 1, statics of results: Repeated measures comparison of planning across LLMs

We evaluated out-of-the-box emergent or native ability of different LLMs on the cognitive map tasks. Table 2 shows the statistical analysis highlighting the contributions of each factor to regression model's fit of LLM model performance. The magnitude of chi-square test statistics indicate contribution to overall model fit. Figure 3 compares the performance of all LLMs across all latent graph structures. Table 3 shows mean and standard error for planning performance across tasks and LLMs.

The results in Table 2 indicate that the model engine ($\chi^2(11) = 689.36$, p < .001.), graph ($\chi^2(11) = 7247.30$, p < .001.), condition ($\chi^2(11) = 1475.03$, p < .001.), and domain ($\chi^2(11) = 304.06$, p < .001.) each yielded significant chi-squared statistics. This means that not only did different LLMs performed differently, but each LLM's performance varied as a result of varying graphs, domains, and conditions. Conversely, the temperature showed a non-significant chi-squared statistic ($\chi^2(11) = 1.01, p = .6022$) and the interaction between the model and temperature was also non-significant ($\chi^2(11) = 4.65, p = .997$). Noteworthy, the interactions among graph-domain, graph-condition, domain-condition, and graph-domain-condition were all significant (all p's < .001). The interactions among graph-domain ($\chi^2(11) = 689.36, p < .001$), graph-condition ($\chi^2(50) = 1392.82, p < .001$), domain-condition ($\chi^2(39) = 524.48, p < .001$), and graph-domain-condition ($\chi^2(108) = 1002.93, p < .001$) were all significant.

In summary, while the 'temperature' and the interaction of 'model' and 'temperature' do not show significant effects in Experiment 1 (but show difference in Experiments 2 and 3), all other factors and their interactions significantly contribute to the variations in the dependent variable. Considering both individual and interactive effects, this finding shows that LLM performance on cognitive map and planning tasks was not robust to the graph structure of the problems, the domain, or the task conditions, and it also varied across models (see Tables 2 and 3 and Figure 3 .