# OpenReview forum: "Evaluating Cognitive Maps and Planning in Large Language Models with CogEval"
_NeurIPS.cc/2023/Conference — NeurIPS 2023 poster_

### Official Review · Reviewer_xxkR · 2023-06-20

**Soundness:** 2 fair
**Presentation:** 1 poor
**Contribution:** 3 good
**Rating:** 5
**Confidence:** 3

**Summary:**

This paper describes a test battery to test for the emergence of cognitive maps and planning abilities in LLM. The tests are based on existing cogsci experiments converted into text prompts. For example, to test for a planning ability, the prompt first describes an apartment layout, and then asks the model to plan a path through several rooms to retrieve something. The authors show that LLM are poor at completing such tasks.

**Strengths:**

I think the adaptation of cogsci planning experiments to text prompts is a great idea. There needs to be more tests like this to map the scope of LLM capabilities.

The authors evaluate multiple prompts from the same generative model and compute statistics of LLM responses. This approach stands in contrast to many arguments for emergent intelligence in LLM, which derive from anecdotal examples. It is important to do actual experiments with LLM.

**Weaknesses:**

This is a presentation issue, but it affects my ability to understand and evaluate the paper. The writing is unedited. There are a lot of unnecessarily long sentences that can be condensed for pragmatics and readability. The grammar and punctuation are weird, like this might be an initial draft. I had to read the same content multiple times, and missed important details throughout the paper.

Figures are odd, seem to be just thrown in together from unrelated bits. For example Figure 1 shows examples of graphs A-E. Are these all the graphs used in Experiment 1? If so, this is not stated. Weirdly, each graph is shown in a different pictorial style. I'm sorry, but this looks like someone just downloaded different types of graphs from google search, panel E has a random red outline around it.

Experiment design is not given. I can not check the statistics without the experiment design. What are these degrees of freedom? It looks like multiple regression models were built? I think the intention is a good one, but it is not clear what was done

**Questions:**

1.
"
We evaluated the robustness of our findings by regenerating the results for each combination of factors
and parameters multiple times and applying a statistical model of how each of these factors contribute
to variance in LLM performance.
"
What does this mean? Where is the exact experimental design? How many trials were simulated? Which conditions were evaluated?

2.
Where is Experiment design for Experiment 1? Where are the graphs?

3.
"198 For the graph community block model, example graphs are shown in Figure 2 ..."
What does this mean? Does Figure 2 show all graphs, or a subset of them? If this is a subset, then how many graphs of each kind were shown?
Where is the "community block model" described?

**Limitations:**

I can not make out what was done exactly, but am happy to read the next draft. The experiments and statistical analysis need to be described so that people could reproduce them.

---

> ### Author Rebuttal · Authors · 2023-08-09
>
> Weaknesses
>
> 1.	Thank you for this thoughtful suggestion. We have edited longer sentences and improved grammar, punctuation, as well as the figures.
> 2.	Yes, all graphs are evaluated in experiment 1, each with 3 instantiations with different spatial and non-spatial domains (18 total environments), 30 measurements of 15 tasks for openAI models and 3 for the remaining 5: a total of 8910 measurements. Domains include spatial ordered rooms (room 1), general spatial (flower field), and social (e.g., Bob is friends with Alice) to explain the structure. The reviewer is correct that we copied the figures of the larger graphs from one of the author’s previous papers. Thanks to the reviewer’s excellent suggestions, we've improved Figure 1 with better graphics.
> 3.	Experiment design: We thank the reviewer for this comment. We describe the experimental design in various portions of the paper but agree the reader would benefit from a concise summary. We elaborate on the design and the statistical approach taken below, and the reviewer can find all the prompts that were used in the prompt viewer link (https://cogeval.github.io/cogmaps/). We are happy to provide further information.
> Our experiment is designed to assess zero-shot LLM behavior on planning tasks. The experimental factors and levels are as follows:
> 1.	Large Language Model (LLM):
> •	OpenAI: GPT-4, GPT-3.5-turbo-175B, davinci-003-175B. Measurements per Factor Combination (MFC): 30
> •	Google: Bard. MFC: 3
> •	HuggingFace: BigScience Bloom-176B. MFC: 3
> •	Cohere: Cohere-52.4B. MFC: 3
> •	Anthropic: Claude-1-52B. MFC: 3
> •	Others: Pythia-20B, LLaMA-13B, Alpaca-7B. MFC: 3
> 2.	Graph structure of the environment: A, B, C, D, E, F
> 3.	Item Domain: spatial (numbered rooms), spatial (spaces), social ties
> 4.	Conditions:
> •	Value-based planning & traversal (what’s the optimal path?)
> •	Reward revaluation
> •	Transition reevaluation
> •	Shortcut (with & without teleportation)
> •	Detour (with & without teleportation)
> •	Temperature: 0, 0.5, 1.0
> It’s essential to note that this is an unbalanced design since the number of measurements differs across LLMs (30 for each of the OpenAI models and 3 per non-openAI LLMs).
> Statistical Analysis: We used logistic regression to model the probability of success as a function of experimental factors. The outcome variable is a binary measure of whether the LLM successfully answered the question or not, given a combination of factors.
> Given the repeated measures, logistic regression effectively accounts for the effect of each factor and their interactions on the probability of success. However, only one logistic regression was run to ensure a comprehensive and unbiased evaluation.
> We hope this clarifies the experimental design and the statistical approach. Please see PDF for new table.
>
> Questions
>
> 1.	We hope our responses above address this concern & are happy to elaborate
>
> 2.	Experimental design: All experiments were designed to test planning ability using different tasks in environments with different underlying structures and domains. All designs begin with explaining an environment (e.g., the rooms and connections in a castle, the social ties among people represented with names) with a Markov Decision Process (MDP) or graph, and rewards. Then a task follows. This task could be “traversal”: prompting the LLM to list the path from a given node to another; the “value path” condition: asking for the optimal path to highest rewards; “reward revaluation”: after responding to value path, the LLM is presented with a second prompt, in which a small local change in the magnitude of rewards is announced and optimal path is probed, to test robustness to changes in rewards; “transition revaluation”: after responding to value path, the LLM is presented with a second prompt with a small local change in the structure of the edges and the optimal path question; “detour”: after responding to value path, the LLM is presented with a second prompt, in which a specific previous path is blocked and the LLM needs to reroute and find a detour using the structure of the environment; “shortcut”: after responding to value path, the LLM is presented with a change in the structure that should reveal a shortcut, so the LLM is prompted for the optimal reward path again to test whether it can identify it.
> 3.	“Where is the "community block model" described?”  The block model is described in the submitted Figure 2.  This is most easily seen looking to the figure to the far right of this image left.  Each block (or community block) contains 5 vertices.  In the image to the right, each separate color here represents its own “community” (there are three of them), where nodes have a specified likelihood of being connected to other nodes in the same block (probability of intra-connection).  In this example image on the right, the likelihood of connection within a block is set to 100%, creating a fully connected clique (all vertices are connected to one another within the block.
> This has now additionally been further clarified in the supplementary material as follows: “To systematically evaluate GPT-4's planning or graph traversal failure modes, we created a three-block community graph structures where each block contains five vertices. Using this approach, we vary the connection density within each community block and ask GPT-4 to perform reasoning tasks over each permutation of the graph structure as block density is varied.
> Example graphs are shown in Figure 2 with the community graphs starting as simple line graphs on the left - representing the sparsest level of connectivity. We then create a new edge within each block for each iteration of the experiment until each community block forms a clique structure as seen on the right of Figure 2”.  For the experiment, we simply vary the connection probability within a block and observe how the LLM performance varies as we move from a largely disconnected block (Figure 2, left) to a fully connected block (Figure 2, right).

---

> > ### Comment · Reviewer_xxkR · 2023-08-22
> > **Thank you for your responce!**
> >
> > I have increased my rating to a 5.

---

### Official Review · Reviewer_Vmb5 · 2023-06-26

**Soundness:** 2 fair
**Presentation:** 1 poor
**Contribution:** 3 good
**Rating:** 5
**Confidence:** 4

**Summary:**

This paper evaluates LLMs on a set of tasks that could be solved by cognitive maps, such as goal-oriented planning, or incorporating shortcuts. The work finds that LLMs generally perform poorly at these tasks, and their performance is affected by features such as graph sparsity.

**Strengths:**

* The paper is admirably thorough with experiments and analyses:
    - Assessing a range of LLMs (including varying parameters such as temperature).
    - Evaluating across a range of different graph structures, task paradigms, etc.
    - Creating new stimuli to avoid dataset contamination.
    - Performing regression analyses to determine how different features contribute to model performance, and describing the failure modes observed.
    - These thorough results and analyses suggest that the conclusions are likely to roughly generalize to some extent, and help the reader to understand which features will affect performance.
* The results are interesting, there are a variety of patterns that could be investigated further.

**Weaknesses:**

My primary concern is that the overall framing of the paper is misleading. In particular, the work is motivated with references to the cognitive and neuroscience literature on cognitive maps. This work performs  draw strong conclusions from its experiments such as "no evidence for understanding cognitive maps or planning." Are conclusions such as these justified?

A key issue in analyzing AI in comparison to human or animal capabilities is determining where a performance failure originates: is it a lack of an underlying capability (such as the ability to form a cognitive map), or a more superficial performance issue? Several recent papers have emphasized this point from a cognitive science perspective, and argued that it is essential to ensure fair comparisons between AI and natural intelligence to draw accurate conclusions (https://www.pnas.org/doi/abs/10.1073/pnas.1905334117; for LLMs specifically see: https://arxiv.org/abs/2210.15303).

In that context, it's worth noting that the animal and human experiments cited involved a great deal more experience before the map was tested than the present experiments do. For example, Tolman's latent learning experiments involved the rats fully exploring the maze for multiple days before they were tested with a food reward at the end (and even then, performance continued to improve well after the rewards were first introduced). Or Schapiro's temporal community structure paper involved half an hour of exposure to transitions from the graph; that is, thousands of transitions from a graph with only 15 nodes. This a much denser sampling of experience than the current LLM experiments afford, and it is quite possible that some degree of repeated experience contributes to the ability of natural intelligences to form a cognitive map. The difference in learning conditions is briefly mentioned to in the limitations, but is quickly dismissed; however, the difference in experimental conditions is a fundamental challenge to concluding that LLMs are failing to form cognitive maps like animals/humans do.

Likewise, it is typical in cognitive science to report comparisons to chance-level performance, and an underlying ability is usually inferred from better-than-chance performance, even if that performance is imperfect. For example, in Tolman & colleague's studies, the rats continued to improve for several days after the reward was introduced (that is, their paths were not optimal on the first test), and the rats were clearly stitching together trajectories that they had observed (since there was only a single route through the maze, everything else lead to dead ends), but we still interpret their performance as showing latent learning. It would be useful to report chance-level performance across all conditions, and, in the case that a model performs better than chance across a broad range of conditions, that would suggest some underlying ability, even if it is imperfect. For example, GPT-4's performance seems reasonably high across most conditions in Table 2 (though certainly imperfect).

In addition, explicit comparisons to human performance on these tasks (presented exactly as they are presented to the language model), would strengthen the claim that language models are failing in a fundamental way that humans or animals would not.

These points seem critical to the overall framing of the paper, and also to much of the discussion. I therefore think the paper would be substantially improved by:
1) providing a more nuanced framing of the very interesting results, that suggests they emphasize some limitations of planning in LLMs, without making overly strong claims such as "no emergent planning" or "no evidence of cognitive maps"
2) Providing explicit comparisons to chance-level (and ideally human) performance to help contextualize the results.

**Questions:**

* In the discussion, what explicit experiment is referred to by "We observe that LLMs do better in problems where the entire trajectories are explicitly available in the text prompts, and they only need to piece together partial changes"?

**Limitations:**

See weaknesses; I believe that the limitations of the experimental paradigm not matching the inspiration are not fully discussed, and that more generally the conclusions are not fully supported by the experiments presented.

---

> ### Author Rebuttal · Authors · 2023-08-09
>
> We thank the reviewer for their thorough engagement with our work and thoughtful notes on the strengths and constructive suggestions.
> Weaknesses:
>
> 1- Framing. We agree that nuanced language is more productive. Our goal was to test zero-shot planning behavior in LLMs. If accepted the camera-ready is “Evaluating Cognitive Maps and Planning in Large Language Models with CogEval”, removing "no emergent planning". We have also introduced nuance in various sections. We referred to “no emergent planning”, given LLM failure modes suggest they don’t “understand” how to use a cognitive map for planning, e.g., they hallucinate edges that don't exist, they fall into loops, or fail to use one-step paths (Supplementary figure 3, now in the main text). We did not intend to create a benchmark to compare against human behavior or chance but to compare planning across LLMs. We evaluate robustness in the face of changes against the baseline of simpler tasks and identify major failure modes.
>
> 2- Comparison to rodent/human experiments: We are thrilled that the reviewer has a deep understanding of cognitive maps and rodent experimental work.
> We'd like to note that our goal here was to test whether LLMs have *zero-shot* planning behavior. While we agree with making the wording more nuanced, please note that our goal was not a comparison to human behavior.
> Moreover, comparison to rodent learning would not be fair given rodents can learn but LLMs with frozen weights can’t learn, and zero-shot planning with language is inherently not comparable to rodent learning.
> Meanwhile, in ongoing studies we are using non-zero-shot approaches to LLMs, in context learning (ICL) & scratchpad/CoT, to improve planning in LLMs. That said, we believe that these studies are not in the scope of the present paper. We acknowledge that investigating all these tasks in human behavior would be a fantastic but separate future contribution.  We hope the reviewer appreciates our response.
>
> 3- Comparison to chance:
> We thank the reviewer for this suggestion. We agree that providing an impression of a chance baseline to LLM performance would be an interesting contribution. However, we believe doing so would be non-trivial given our specific design.
>
> 3.1. defining chance is non-trivial: The examples the reviewer provided rely on binary & multiple-choice responses, where defining chance at 50% makes sense. However, it is less clear how to define chance level when we have asked for a trajectory to optimal reward. If we had used multiple-choice responses, we could quantify random chance as 1/k where k was the number of choices. We pose open questions to the LLMs and evaluate their responses. It is not clear how to enumerate a sample space of possible answers.
> However, we believe that comparison of the same LLM across graphs and domains, as well as the comparison of different LLMs' performances on the same tasks/graphs/domains offer ample novel contributions to the field and satisfy cognitive- and neuro- science standards (some authors have worked in cog sci for decades). However, we have considered a number of possibilities below in hopes to align the reviewer’s thoughts with our thinking process on the non-triviality of a chance level.
> Consider traversal from a given node to a destination node. For given a graph, one possibility is to calculate all pairs of “shortest paths”, but this reduces down to the algorithm used for betweenness centrality (which is n^2 * (log n) to compute) and it would produce an exponential number of possible paths. This means that the likelihood of randomly choosing the correct path will almost always be zero, which makes the use of random chance less meaningful.
>
> 3.2. We could also use a random walk algorithm with the following constraints:
> •	Randomly walk through the graph and report a success if the goal node is reached, failure otherwise
> •	No backtracking & no revisiting nodes.  If a previously visited node is encountered again in the random walk, terminate walk as a failure.
> •	Apply uniform transition probabilities based on what transitions are permitted in each condition.
> •	Run repeated random walks, count the proportion of successes as a Monte Carlo estimator of the probability of success by random chance.
> To illustrate, for the value-based planning conditions and transition revaluation conditions on graph A, a random walker that didn’t backtrack would have a *50% chance* of achieving the goal (the room with the most reward). However, once we introduce the shortcut/detour/teleportation conditions, the state graph becomes complicated. The simulation would become more complex for more complex graphs but this random walk Monte Carlo estimation approach for random success probability may be a sound baseline for evaluating LLM performance.
>
> Questions: We appreciate the opportunity to expand on this.
> In smaller graphs, the prompt already expands all the possible paths or trajectories. When there’s a change in the rewards or transition structure, the LLM only needs to change one thing in an already laid out path. However, in more clustered graphs only the one-step connections are laid out in the prompt, but not all paths or trajectories between any given two nodes. This means that the LLM needs to use the transition structure to unroll the trajectories and find the correct path, which is closer to the notion of planning in model-based RL and in cognitive science.
> An observation that speaks to this is that performance on larger graphs is far worse than the smaller ones. It is not just the graph size, LLMs often perform worse on the graph with 15 nodes and 3 dense clusters compared to the 16-node (4-cluster) graph that has more nodes, but better cross-cluster connectivity. The difference: there are fewer paths among clusters in the Schapiro graph, making “planning” more relevant here. The difference is robust to prompt variation. Taken together, these findings support the claim and we are happy to discuss further.

---

> > ### Comment · Reviewer_Vmb5 · 2023-08-11
> > **Thanks for the improvements, and some follow up thoughts**
> >
> > Thanks to the authors for their thoughtful response. I've updated my score accordingly. Some follow-up thoughts below:
> >
> > 1. I believe this reframing will improve the paper.
> >
> > 2. In the context of the above reframing, this issue will likely be improved. However, I have some lingering concerns about the comparison, that depend on precisely how the paper is reframed. For example, if the authors still reference rodent + human studies in motivating the cognitive maps (which I think is a good thing!), then it would still be useful to highlight the distinctions between the experimental methods in that work (e.g. substantial experience, learning as the authors point out) and the zero-shot methods in this work. I understand that the goal is not to make direct comparisons to human behavior, but I think it will be hard to write a paper talking about cognitive maps and citing the prior literature without (implicitly, at least) drawing that comparison. Thus, I hope the authors will highlight these discrepancies in the paper.
> >
> > 3. I agree that chance level performance can be tricky to determine. Nevertheless, I think that including some such comparisons would help to situate the results in context.
> >   - From my perspective, any chance metric that takes account of the graph structure (such as random walks on the graph) is not really the most appropriate chance-level comparison for these experiments, because it effectively presumes some kind of cognitive map (that is, if the model sampled from these distributions, it would thereby be fully respecting the constraints imposed by the graph structure). In a fully no-cognitive-map baseline I would expect chance to be sampling paths truly at random from the set of possible nodes, without respect to any spatial constraints (perhaps sampling without replacement, so that the set of paths is finite).
> >   - Alternatively, the authors might suggest that recognizing pairwise constraints would be possible just from the experienced paths, and so an additional chance level baseline would be sampling from transitions observed in the prompt.
> >   - I think comparing the models' ability to respect local dependecies to the above two chance level baselines would help to elucidate the extent to which the issues the model is facing stem from lack of *any* cognitive map (i.e. not respecting local dependency constraints, or only respecting observed ones) vs. inability to plan over longer distances.  I think that such a comparison would help to quantitatively clarify some of the qualitative claims in the current discussion.
> >   - I do think it would be reasonable to also compare to a random walk over the graph with no repeats baseline; that would be more of a "cognitive-map-but-no-planning" comparison, which would also be useful (but I'd see the above as more valuable).
> >
> > Questions. Thanks, this clarifies things.

---

> > > ### Author Response · Authors · 2023-08-19
> > > **Thank you and following up**
> > >
> > > 1- We appreciate it.
> > >
> > > 2- We agree with the reviewer that the difference between human experiments and our prompts for LLMs is important. We already noted this in the submitted manuscript under limitations, lines 298-304:
> > > “in the human experiments that influenced our prompts, participants learn gradually, experiencing states one-by-one, but were only tested after they showed signs of learning, similar to a model-based RL agent having the transition structure and using it for inference and planning. To address this difference, we present the environment’s structure in linguistic format. In all cases, the participant or model had to identify the goal location based on instructions and infer the policy towards the goal, which is the room with the maximum reward.”
> > > Second, on lines 318-319 we explicitly said that we use a functionalist notion of cognitive maps and planning and not a “human-like” notion:
> > > “we evaluated emergent cognitive capacities in LLMs in a functionalist and multiple-realizablity sense rather than requiring any assumptions of them being "human-like" [31]”
> > > Even Tollman’s original 1948 paper imbues cognitive maps with far more abstract & general intentions. Behrens et al. 2018's “What is a cognitive map?” defines cognitive maps as relational structures of knowledge, & Epstein et al. 2017 as “a unified representation of the environment to support memory and guide future action”. Consistently, we functionally test whether LLM behavior in planning tasks is consistent with having a unified representation of environment that can be accurately recalled for planning.
> > >
> > > We described the structure of the env & asked questions to test whether LLMs can *extract a unified representation (cognitive map) and accurately recall it for flexible planning* & found that while some LLMs can list state-state transitions, when it comes to using this knowledge for planning they *hallucinate edges that don’t exist* (GPT-4: 25.57% responses on Schapiro graph were wrong due to hallucinations), *fall into loops*, or say irrelevant things. In a functionalist spirit we a) report this performance & classify failure modes, b) suggest that these failures are inconsistent with LLMs accurately using a unified representation of the environment (cognitive map) for planning. We agree with the reviewer’s broad comments & we've touched on them in the MS but can further clarify.
> > >
> > > 3- We appreciate the reviewer noting sampling from observed transitions in the prompt to check if they’re from the set. As shown in Supplementary Figure 2, now in the main MS, we visualize 3 failure modes of GPT-4. Among them hallucinating edges that don’t exist speaks to this. Namely, for Schapiro graph 25.57% of GPT-4 responses were incorrect because of hallucination. This is despite being able to list the tuples when asked as opposed to planning tasks. This speaks to a failure of “a unified representation of the environment for memory & planning”, or cognitive map, revealing planning failure due to inaccurate recall of transition structures. The reviewer’s point is thus relevant & helpful: “not respecting local dependency constraints” and inspired us to compute the %hallucination. If the reviewer agrees, we will add this to the paper.
> > >
> > > We kindly note that many studies in the cognitive map literature do not use a chance level but a task condition as baseline (Garvert et al 2017; Momennejad et al 2017). Even Tolman & many rodent experiments don’t focus on chance levels but a condition as baseline (e.g., latent learning contrasts having vs. not having explored the environment before rewards are introduced). Similar to Tolman’s latent learning, a given LLM or condition can serve as a baseline in statistical analyses to Compare:
> > > A) tasks with on (traversal) as baseline
> > > B) LLMs (odds of success are 6.1464X higher for GPT-4 > alpaca-7b)
> > > C) robustness to domain (baseline: spatial)
> > > D) robustness to local changes (reward reval, detour, etc) & temperatures
> > > Such analyses revealed how different factors (model engine, graph type, domain, task) & their interactions affect the odds of success for LLMs on each task.
> > >
> > > We agree with the reviewer that random walks do not offer a satisfactory chance level. We believe that given the open multistep responses our design is not comparable to classification or experiments that require chance. In our opinion, artificially imposing nontrivial chance metrics might unfairly bias the interpretation to make it seem like LLMs do better. Moreover, LLMs are explicitly trained on next word prediction so randomly sampling any tuples does not seem like an appropriate baseline. Thus, we respectfully do not believe that random sampling is a fair comparison.
> > >
> > > We hope our responses, rooted in over a decade of experience with cognitive maps, have addressed the reviewer’s concerns & are happy to clarify further. We hope our difference in optimism about LLM abilities does not impede the reviewer from raising the score!

---

> > > > ### Comment · Reviewer_Vmb5 · 2023-08-20
> > > > **Thanks, I would still prefer some more interpretable comparisons**
> > > >
> > > > Thanks for continuing to follow up. I think this most recent response has helped me to achieve a bit more clarity out of what I believe would improve the paper.
> > > >
> > > > I think fundamentally the issue is that performance on these tasks is hard to interpret without comparison to some baseline that is a "known quantity" that the reader will understand, and that supports the argument the authors are trying to make. That is, just as we measure distance with respect to some baseline unit, it is useful to contextualize performance on a task with respect to some well-understood comparison. While chance level is one such comparison (and RL research in AI often includes random baselines, even for multistep tasks), there are other approaches such as comparing to another well-known system or model. I will point to several approaches to this below.
> > > >
> > > > 1) The Tolman studies indeed use no-exposure as a baseline condition. This is a reasonably interpretable no-prior-experience baseline; and we interpret the rats as learning because they outperform it given prior experience. That is, this comparison supports Tolman's argument that latent learning occurs. I would be satisfied with such a comparison in this work; it would effectively fulfill the goal of what the chance level baseline was supposed to do, which is to provide a comparison of whether some sort of meaningful learning occurs. However, to my understanding the authors did not perform a no-exposure comparison (please correct me if I have misunderstood however).
> > > >
> > > > 2) many more recent computational studies make explicit comparisons to well-understood models. For example, if I recall correctly Momennejad et al 2017 compared human performance to a model-free baseline, a model-based planner, and SR. By making these comparisons to explicit computational models (in addition to the task condition comparisons), the authors more clearly highlighted what features humans seemed to use to solve their tasks. Again, these models provide benchmarks the reader can understand when assessing the performance of the system in question. Likewise, I think such baselines would be helpful for the corresponding tasks in this work; and more generally, comparing to well-understood models would help to provide a clearer benchmark for model performance.
> > > >
> > > > 3) alternatively, as I suggested originally, explicitly comparing to human performance can provide such a "known-quantity" baseline in some sense; while we don't understand human cognition in all its detail, it at least provides a familiar yardstick by which a system can be measured, and is a common standard in ML.
> > > >
> > > > By contrast, comparing to another LM or even another task condition is harder to interpret; there are many factors that could affect performance in each condition, so it's not entirely clear what the comparison means, other than to show that one model is better or one graph is harder. This issue is exacerbated because the comparisons are not clearly presented, at least in the original draft of the paper. For example, if I were interested in transition revaluation in Table 2, which other condition would be relevant to compare it to in order to isolate this ability? And even if this were clarified, the core point remains: these comparisons don't seem to communicate how the reader should understand the performance of the system overall.
> > > >
> > > > The authors note that "artificially imposing nontrivial chance metrics might unfairly bias the interpretation to make it seem like LLMs do better," and while I can understand this concern, I'm not sure that I agree it would bias the interpretation any more than leaving it out does. As highlighted in our discussion above, there are many versions of randomly acting baselines that could be used to compare different types of random behavior. What, precisely, is it that the authors think the models are doing that is better than these but doesn't count as some kind of real-but-imperfect planning? The authors could of course contextualize such results through many of the methods discussed above, such as comparisons to a well understood baseline model (such as a model-based planner, which presumably would be much better in many conditions), condition comparisons (e.g. "models only perform above random transition sampling on paths that have property X" or "as graphs get more complex, models get closer to chance"), or other metrics like quantifying % hallucinations (which I agree would be useful).
> > > >
> > > > In summary, I feel the current results are a bit hard to interpret without comparison to systems that can be more readily understood.
> > > >
> > > > The authors are, of course, also welcome to ignore these comments; I will re-emphasize that I find the experiments useful, and I have already increased my score to an accept (if a borderline one) in response to the reframing and improvements in the original response. Given that several other reviewers have given higher scores the paper may well be accepted regardless.

---

> > > > > ### Author Response · Authors · 2023-08-21
> > > > > **Thank you and final comment**
> > > > >
> > > > > We sincerely thank the reviewer for their comments and engagement. We would kindly note the following as our final response:
> > > > >
> > > > > Re the following comments:
> > > > > "I think fundamentally the issue is that performance on these tasks is hard to interpret without comparison to some baseline that is a "known quantity" that the reader will understand, and that supports the argument the authors are trying to make. "
> > > > >  " I feel the current results are a bit hard to interpret without comparison to systems that can be more readily understood."
> > > > >
> > > > > While we agree that comparison to other models is relevant (see 2 below), since model-based or SR agents would not hallucinate transitions that don't exist 25% of the times, we respectfully disagree that the results are hard to interpret. It seems clear that LLMs are failing at accurate recall and use of experienced edges for planning, and failing at having a unified representation of the environment that can be unrolled for planning paths that have not been directly spelled out. This is consistent with previous observations reported on LLM failures (e.g., see LLMs still can't plan, or work by Melanie Mitchell, infamous cases of failures due to hallucination, or reported failures in causal inference).
> > > > > Detailed responses below.
> > > > >
> > > > > 1- Traversal and value-based tasks are here used as the baseline task to compare, when something in the environment hasn't changed. If the reviewer deems necessary, we can add a condition where there's a second prompt rather than just the first prompt to test a "no change" condition, if what you're asking is to be comparable to the 2-prompt nature of conditions with local changes.
> > > > >
> > > > > Here's an example of what using a task or LLM as baseline would look like. Using the smallest LLM, i.e., replicate-alpaca-7b, as the baseline, the odds of gpt-4-32k success are approximately or 6.1464 times higher than the baseline, holding other factors constant. The same can be done for tasks, e.g., using traversal as baseline.
> > > > >
> > > > > 2- We agree about the relevance of comparison with other models, which is the reason we cite and mention this directly in the paper. Moreover, please note that a model-based or successor representation behavioral policy (compared in Momennejad et al. 2017) would never yield a 25% hallucination using tuples that were never experienced.
> > > > >
> > > > > So, we respectfully disagree that the results are hard to interpret. However, we understand that the results are not aligned with the reviewer's optimism about LLMs and respect the reviewer's different opinion. That said, we agree with the reviewer that comparison to other models, e.g., RL or dynamic programming, is helpful. We have already mentioned them in the paper, but we can certainly highlight it more or run them for the specific prompts. We thank the reviewer for the opportunity to spell this out. As noted earlier, we can compare the different LLMs too: using the smallest LLM, i.e., replicate-alpaca-7b, as the baseline, the odds of gpt-4-32k success are approximately or 6.1464 times higher than the baseline, holding other factors constant.
> > > > >
> > > > > 3- We thank the reviewer for this idea. We believe that direct comparison to human behavior is another worthwhile project that is outside the scope of this one paper (which we can barely fit in the 9-page limit of neurips).
> > > > >
> > > > > Overall, we are very grateful that the reviewer engaged deeply with our work. We can certainly clarify the baselines and statistical analysis based on the baseline, & comparison to other models. We agree that this is a novel approach, and it might be hard to classify it in a category of previous work comparing models to human behavior. We believe that as a part of a larger research program, this work can stand on its own - and then some, given the wide comparison with multiple LLMs. We believe reporting the present results is the responsible scientific duty, while future directions are both scientifically interesting and worthy. We thank the reviewer again for their kind engagement, which has allowed us to deepen our clarification of the work. We're happy to lengthen the sections on comparison to MBRL or SR agents and future directions.
> > > > >
> > > > > Thanks again!

---

### Official Review · Reviewer_ydCJ · 2023-06-29

**Soundness:** 3 good
**Presentation:** 4 excellent
**Contribution:** 4 excellent
**Rating:** 7
**Confidence:** 4

**Summary:**

The paper presents CogEval, a set of best practices ported from cognitive science on how to do behavioral evaluations. The authors also transcribe new tasks from human reinforcement learning and planning into text, such that LLMs can be tested on them. On these tasks, the authors do not observe evidence for an emergent capability for planning in large language models.

**Strengths:**

- The presentation of CogEval is clear and potentially useful for the ML community. I’m excited to use CogEval in my own work.
- Thorough and thoughtful discussion.
- Exhaustive experimental setup—a feature perhaps enabled by the principled CogEval framework!
- The paper is lucid overall and easy to read.

**Weaknesses:**

There’s some mild overselling in the paper, given the empirical results, which only demonstrate the failure of an existence proof.  Defining the conditions under which we can declare definitively that there is no emergent planning. However, given that we have not rigorously defined these conditions, for example, “No Emergent Planning” in the title seems too strong.


**Questions:**

- Are you planning to release these tasks transcribed to text as a new testbed?
- Does transcribing these tasks to text make them harder? Perhaps a vision-and-language model like GPT-4 or Flamingo might perform better on pictures of the task, in which case, perhaps your conclusions would be different.
- Why is “Measurement and Evaluation for Large Language Models” capitalized in line 6?

Nits:
- Line 14: Stay in active voice here for consistency?
- Figure 3: Can you make this less blurry?

**Limitations:**

The authors have adequately addressed limitations.

---

> ### Author Rebuttal · Authors · 2023-08-09
>
> We are deeply grateful that the reviewer finds CogEval clear and thorough and are delighted to read they may potentially try it or use it in their work. We hope that we have addressed their helpful and constructive suggestions below and are more than happy to address any further feedback.
>
> Weaknesses:
> •	We appreciate the reviewer’s suggestion and find it a fair assessment. In the camera-ready version, we plan to remove “no emergent planning” from the title. We have already begun editing the text to make the conclusions more nuanced in various sections. Our interim new title is “Evaluating Cognitive Maps and Planning in Large Language Models with CogEval”.
>
> Questions:
> •	All the prompts are already available in the anonymous link we provided for the reviewers.
> •	•	This is an excellent question, and we hope very much to test this in the future. Given our multiple tests and checks we know that GPT4 can extract the correct tuples from the instructions, both spatial and social. However, GPT4 fails at using these same tuples during planning with three main failure modes, please see Supplementary figure 2, also included in the rebuttal PDF. If accepted, this figure will be moved to the main manuscript. These failure modes include 1) hallucinating edges that are non-existent, 2) taking unnecessary moves that unnecessarily lengthen the path, even when there’s a 1-step transition available that GPT4 has captured from the description, and 3) falling into loops. The question of the influence of a visual LFM on each and all these failure modes is surely of interest. Would visual reasoning improve all, some, or worsen? We think pursuing the reviewer's suggestions could be an excellent follow-up study.
> •	The reviewer is correct, this is inherited from an older abbreviation prior to CogEval. We have fixed it per the reviewer’s suggestion.

---

> > ### Comment · Reviewer_ydCJ · 2023-08-14
> >
> > Thanks to the authors for acting on my few suggestions. The authors more accurately represent their results.

---

### Official Review · Reviewer_UmDW · 2023-07-07

**Soundness:** 3 good
**Presentation:** 2 fair
**Contribution:** 3 good
**Rating:** 7
**Confidence:** 3

**Summary:**

This paper proposes an evaluation of large language models with respect to their ability to solve problems that require use of latent cognitive maps. Evaluation focuses on different underlying graph structures, and the influence of chain-of-thought inference on performance.

**Strengths:**

* I really liked the very clear outline of the motivation of the research design in the introduction of Section 2.
* The experiments are extremely thorough, and the goal of designing experiments with statistical robustness in mind is good.
* Good discussion around the capabilities and limitations of LLMs

**Weaknesses:**

Some of the presentation could be refined:
* The description of Figure 1 is somewhat difficult to understand. References to future aspects of the paper (e.g., "Experiment 3") are undefined, which makes it more difficult to understand.
* The figures / tables should appear closer to where they are referenced in the text.
* I'd suggest reordering 2.1, so that the tasks (i.e., maze learning) are described before the experimental setup.
* Some details could use more context. e.g., what is temperature? What are the graph structures A/B/C... etc?
* The discussion on BFS/DFS prompting should be moved to the experimental setup section (2.3). I also didn't quite understand the distinction between these two; an example would help.
* Formatting of Figure 3 can be improved
* What is "dialogue" referring to throughout the paper? I don't believe an actual multi-turn dialogue is taking place during the evaluation
* Text of Table 2 is really tiny

**Questions:**

* I am not completely familiar with the term "cognitive maps", and I'm not sure if this more of a metaphor, or usually applied to actual spatial reasoning tasks (hence "map"). Since the examples in the paper are about navigation, I am wondering if this means the domains studied are mostly about literal maps, or if there are other domains usually studied with the framework of cognitive maps. Are there other domains covered in this study? If so, what are they?
* If there are different domains, how does performance vary across them?

**Limitations:**

Yes

---

> ### Author Rebuttal · Authors · 2023-08-09
>
> We sincerely thank the reviewer for their positive evaluation of our work as well as their careful and constructive questions and suggestions. Please see our responses below. We hope that we have addressed any concerns and are happy to address further questions in the discussion period as well.
>
> Weaknesses:
>
> 1.	We have now improved Figure 1 caption for clarity.
> 2.	Great suggestion, we reorganized the figures and hope to show it in the camera ready.
> 3.	Appreciated, we have now accommodated this.
> 4.	Temperature in LLMs determines randomness in the generated response through the softmax function. This manipulates the probabilities of the next word in a sequence. Because of this, temperature can be thought of as a parameter controlling the diversity of the output. When temperature is set to 0, this results in deterministic or greedy responses with less variance (Note: OpenAI has made it known that even temperature=0 is not entirely deterministic, though this is as close as you can get). Higher temperatures, especially closer to 1, create more diverse and varied text upon repetition. While this is helpful for tasks that may require varied responses or creativity, it’s not great for responses that require precision such as planning trajectories. We are more than happy to add this to the supplementary or integrate a summary in the main text if the reviewer sees fit.
> The graph structures are the latent structures of the problems discussed in the paper.
> 5.	Thank you. Breadth First Search and Depth First Search instructions are provided in the supplementary (lines 55-75), and we have pasted them below to address the reviewer’s question.
>
> 5.1.	BFS (Breadth First Search) instruction:
> “Think carefully before you respond. You can try using Breadth-first search (BFS), it is a graph traversal algorithm that visits all the vertices of a graph in breadth-first order, starting from a given source vertex. In BFS, vertices are visited in layers, where the vertices at distance 1 from the source vertex are visited first, followed by the vertices at distance 2, and so on. BFS uses a queue data structure to keep track of the vertices to be visited, and it ensures that no vertex is visited more than once. BFS is useful for finding the shortest path between two vertices in an unweighted graph, or for exploring all the vertices in a graph.”
>
> 5.2.	DFS (Depth First Search) instruction:
> “Think carefully before you respond. You can try using Depth-first search (DFS), it is a graph traversal algorithm that visits all the vertices of a graph in depth-first order, starting from a given source vertex. In DFS, the algorithm traverses as far as possible along each branch before backtracking. DFS uses a stack data structure to keep track of the vertices to be visited, and it ensures that all vertices connected to a visited vertex are explored before backtracking. DFS is useful for finding cycles in a graph, for exploring all the vertices in a graph, or for finding a path between two vertices. However, unlike BFS, DFS does not guarantee that the shortest path is found.”
>
> 6.	We agree, and have updated Figure 3 accordingly, which you can find in the PDF. Additionally, we have audited some of the prompts and can provide further explanation as needed. All prompts can be found in the interactive tool with the prompts Chatbot Visualization (cogeval.github.io).
> 7.	While we don’t use the term "dialogue" in the manuscript, in tasks that test robustness of LLMs to changes in rewards or transition structures, there is a 2-step question. Following Momennejad et al. 2017, 2018, First, the graph is explained and the LLM is probed for the optimal policy, then a partial change is described and the LLMs is probed a second time for the optimal policy. The correct response to the second question requires the integration of information in the first and the second prompts.
> 8.	We have now changed the table size.
> Thanks again for the helpful suggestions, and we are more than happy to address any further feedback.
>
> Questions:
>
> 1.	Thank you for the question. In lines 39-51 in the original manuscript, we discuss what a cognitive map is and in 52-62 we explain why LLMs would show that capacity. Briefly, the term was coined in cognitive science by Tolman’s 1948, where he reviewed decades of planning and navigation research using mazes. It refers to the relational structures that are stored in memory, representing a map of the state-space (which can be spatial or non-spatial, e.g., social relations) that is held in the mind rather than externally, hence the phrase "cognitive map". The specific structure of this map, whether it is one-step, multi-step, multi-scale, etc. has all been a matter of research over the past century. Since Tolman, the neural underpinnings of cognitive maps won a Nobel prize and as Tolman intended it, many studies discuss cognitive maps in terms of non-spatial maps (e.g., social maps, associative relational structures) as well. Many models have been proposed, with representation learning and RL being of special relevance, including debates over whether the map is Euclidean or not. We are happy to clarify further if lines 39-62 (and references 5, 1, 9) are unclear or add further sections in the supplementary.
> 2.	Thank you for this helpful question. We have now provided a figure to address this constructive question in the attached PDF.

---

> > ### Comment · Reviewer_UmDW · 2023-08-21
> >
> > Thank you for your rebuttal and apologies for not responding earlier. It has answered my questions and I would still like to see this paper accepted.

---

### Author Rebuttal · Authors · 2023-08-10

Dear reviewers,

We are deeply grateful for your careful and detailed review of our work as well as your constructive questions and suggestions. Given the space constraints we have tried to address your questions as best as we could and have provided further material in the *attached PDF*. We hope that we have addressed all your comments and are more than happy to engage in further discussion. We hope to be able to show you how these comments have changed the paper overall in the camera-ready version.

Thank you,

The authors

---

### Decision · Program_Chairs · 2023-09-21

**Decision:**

Accept (poster)

**Comment:**

This paper presents CogEval, a cognitive science-inspired protocol for evaluating language models. All reviewers unanimously found the experiments thorough across a variety of dimensions, and appreciated the statistical rigor of the analyses. The discussion has been very productive. Thanks to the reviewers’ excellent comments and deep engagement, authors have improved their framing and added more nuance to the interpretation of the results. They have also made attempts at clarifying details regarding the experimental setup and statistical analysis, as well as presentation of the figures/tables.

However, the main remaining concern is the following: The proposed tasks can be solved with cognitive maps, and the LMs are not able to solve them perfectly. However, equating this with “no emergent planning” is an incorrect interpretation. Several reviewers argue that it is an oversimplification and a stronger conclusion than the observations suggest. While the authors’ proposed edits address this concern partially, they have not provided further contextualization to other recommended baselines that best resemble a “chance” control setup and might provide a fairer testbed for comparison.

In short, the experimental setup is laudable, and the paper has resulted in engaging discussions. Facilitating such discussions more broadly could be beneficial for the community. In any case, I strongly recommend authors consider the insightful suggestions by the reviewer regarding alternative control setups to better detect "imperfect, yet better than chance" planning for future work, and highlight that as a limitation of the current draft.